Manuscript prepared for Geosci. Model Dev.
with version 2015/04/24 7.83 Copernicus papers of the LATEX class copernicus.cls.
Date: 25 October 2016

# Advantages of using a fast urban boundary layer model as compared to a full mesoscale model to simulate the urban heat island of Barcelona

M. García-Díez [*1], D. Lauwaet[2], H. Hooyberghs[2], J. Ballester[1], K. De Ridder[2], and X. Rodó[1, 3]

[1]Institut Català de Ciències del Clima, Barcelona, Catalonia, Spain
[2]VITO, Antwerp, Belgium
[3]Institució Catalana de Recerca i estudis Avançats (ICREA), Barcelona

*Correspondence to:* Markel García-Díez (markel.garcia@ic3.cat)

**Abstract.** As most of the population lives in urban environments, the simulation of the urban climate has become a key problem in the framework of the climate change impact assessment. However, the high computational power required by high resolution (sub-kilometer) fully coupled land-atmosphere simulations using urban canopy parameterizations is a severe limitation. Here we present a study on the performance of UrbClim, an urban boundary layer model designed to be several orders of magnitude faster than a full-fledged mesoscale model. The simulations are evaluated with station data and land surface temperature observations from satellites, focusing on the Urban Heat Island (UHI). To explore the advantages of using a simple model like UrbClim, the results are compared with a simulation carried out with a state-of-the-art mesoscale model, the Weather Research and Forecasting model, which includes an urban canopy model. This comparison is performed with driving data from ERA-Interim reanalysis data (70 km). In addition, the effect of using driving data from a higher resolution forecast model (15 km) is explored in the case of UrbClim. The results show that the performance on reproducing the average UHI in the simple model is generally comparable to the one in the mesoscale model when driven with reanalysis data (70 km). However, the simple model needs higher resolution data from the forecast model (15 km) to correctly reproduce the variability of the UHI at a daily scale, which is related to the wind speed. This lack of accuracy on reproducing the wind speed, especially the sea breeze daily cycle, which is strong in Barcelona, causes also a warm bias in the reanalysis driven UrbClim run. We conclude that medium-complexity models as UrbClim are a suitable tool to simulate the urban climate, but that they are sensitive to the ability of the input data to represent the local wind regime. UrbClim is a well suited model for impact and adaptation studies at a city scale without high computing requirements, but does not replace the need for mesoscale atmospheric models when the focus is on the two-way interactions between the city and the atmosphere

---

*markel.garcia@ic3.cat

# 1 Introduction

According to the United Nations, more than 50% of the world population lives in cities, and this percentage is expected to increase in the coming decades. The urban environment is known to modify the local climate in several different ways. The most well-known is the so-called Urban Heat Island (UHI) effect, that consists on temperatures being several degrees higher over the urban area with respect to its rural surroundings. Due to anthropogenic climate change, the frequency of heat waves is 30 expected to undergo a widespread increase (Meehl and Tebaldi, 2004) in the following decades. This raises concerns about the vulnerability of people living in urban areas. Although the magnitude of the UHI effect does not necessarily increase due to global warming (Lauwaet et al., 2015), it has been shown to be large enough to pose significant impacts. The most important ones are human health, through heat stress (Gabriel and Endlicher, 2011; Dousset et al., 2011) and energy consumption 35 (Kikegawa et al., 2006; Kolokotroni et al., 2012).

The physical causes of the UHI effect were enumerated by Oke (1982), but the relative contribution of each one is still discussed. Zhao et al. (2014) used satellite observations and a model simulation to calculate the relative contribution of the different causes of the UHI in 65 cities of North America. They considered contributions from modifications in the radiative balance, evapora-40 tion, convection efficiency, heat storage and anthropogenic heat sources. They found that the relative contribution of these factors depends on the local background climate of the city and on the time of the day. In general, during daytime, convection efficiency and evapotranspiration are the main drivers of the UHI, while heat storage is the most relevant during the night. Zhao et al. (2014) used satellite retrieved land surface temperatures, but these can differ from screen level temperatures. Others have 45 highlighted the complexity of the problem of measuring the UHI, given that it is difficult to monitor the urban climate with enough detail and reliability (Arnfield, 2003). Furthermore, the complexity of the urban surface, featuring anisotropy and vertical surfaces, makes it complicated to sample by satellites (Voogt and Oke, 1998). These difficulties with the observations increase the value of numerical simulations, that can produce detailed fields which are not observable. At the same time, the 50 lack of observations hampers the evaluation of these simulations.

Recently, urban canopy models or parameterizations have been included in many Regional Climate Models (RCMs) (see. for example Huszar et al. (2014) and Ching (2013)). RCMs are limited area models used, among many other application, to dynamically downscale climate change projections from coarse resolution Global Circulation Models. Nevertheless, computational power limita-55 tions do not allow RCMs to reach the level of resolution that is required to resolve most of the cities. Here we study, by using a simplified model that only accounts for the Planetary Boundary Layer (PBL) and the surface physics, the possibility to reach resolutions of 250 m with affordable computational resources. This model, called UrbClim, has been developed by De Ridder et al. (2015), and is described in section 2.3. UrbClim has been already evaluated in a few European cities (De Ridder 60 et al., 2015; Zhou et al., 2015; Lauwaet et al., 2016) and used to generate climate change projections

(Lauwaet et al., 2015). Note that UrbClim has a different, more specific scope than the RCMs, being focused on the fast and computationally light simulation of the UHI and the heat stress in the urban environment, so that it can be easily transferred between cities. While this scope covers many applications, it must be mentioned that RCMs are required to reproduce the two-way interaction between the city and the atmosphere affecting the rain, storm initiation and other phenomena, at the expense of a much larger computational cost.

Statistical downscaling can also be considered as an alternative (os complementary) methodology to asses the urban climate. However, statistical downscaling is observation-dependent, and spatially detailed observations at a city scale covering long periods are very rare. UrbClim, instead, allows for long multi-decadal adaptation experiments where changes in the city surface parameters can be tested (e.g., the color of the roofs or the evaluation of the effects that changes in construction materials in building may have). The main aim of this study simulate only the fundamental processes that cause the UHI, so the model is lightweight, but still based on physics, and thus allowing sensitivity experiments to be conducted.

In the present paper, we evaluate UrbClim over the city of Barcelona, and compare it with a standard mesoscale model, the Weather Research and Forecast model (WRF), using a Urban Canopy parameterization. Namely, the Single Layear Urban Canopy Model (SLUCM) was used, which was developed by Kusaka et al. (2001) and coupled to WRF by Chen et al. (2011). It has been verified in several studies (Lee et al., 2011), and used for future climate change projections (Argüeso et al., 2015; Georgescu et al., 2014; Kusaka et al., 2012).

Barcelona is located in the Euro-Mediterranean region, which has been defined as a primary climate change hot spot (Giorgi, 2006), as it emerges as an especially responsive area to climate change, with more frequent, longer and harsher summer heat waves (Meehl and Tebaldi, 2004; Ballester et al., 2009, 2010a, b). Taking into account that the Mediterranean countries are currently more vulnerable to environmental summer conditions than other European societies, the larger magnitude of the projected temperature increase is expected to become a major challenge for public health in summer (Ostro et al., 2012). For example, the negative effects of the record-breaking 2003 heat wave in central and southern Europe were particularly damaging in the Euro-Mediterranean arch (Robine et al., 2008). The seasonal mortality excesses were indeed similar in Spain (13.7%), France (11.8%) and Italy (11.6%), although temperature anomalies were twice larger in France than in the southern countries (Ballester et al., 2011). This larger sensitivity to environmental conditions is exacerbated by urban pollution especially in old people living in cities with pre-existing or chronic cardiovascular and respiratory diseases (McMichael et al., 2006). Taking into account all these considerations, the city of Barcelona emerges as a particularly vulnerable area within the continent. Barcelona is located in northeastern Spain, surrounded by the Mediterranean Sea in the south and east, a small 500 m mountain range in the northwest, and two rivers in the southwest and northeast (figure 1, and see also figure 2 for the location of the urbanized area). Its Mediterranean climate (Csa in the Köp-

**Figure 1.** a) Topography of the UrbClim domain and locations of the meteorological stations. The two stations used as a reference of the urban and rural climates are highlighted with red stars. b) Three WRF domain edges (red squares) and UrbClim domain edges (black contour), together with the topography of the WRF domains with 10, 3.3 and 1.1 km resolution.

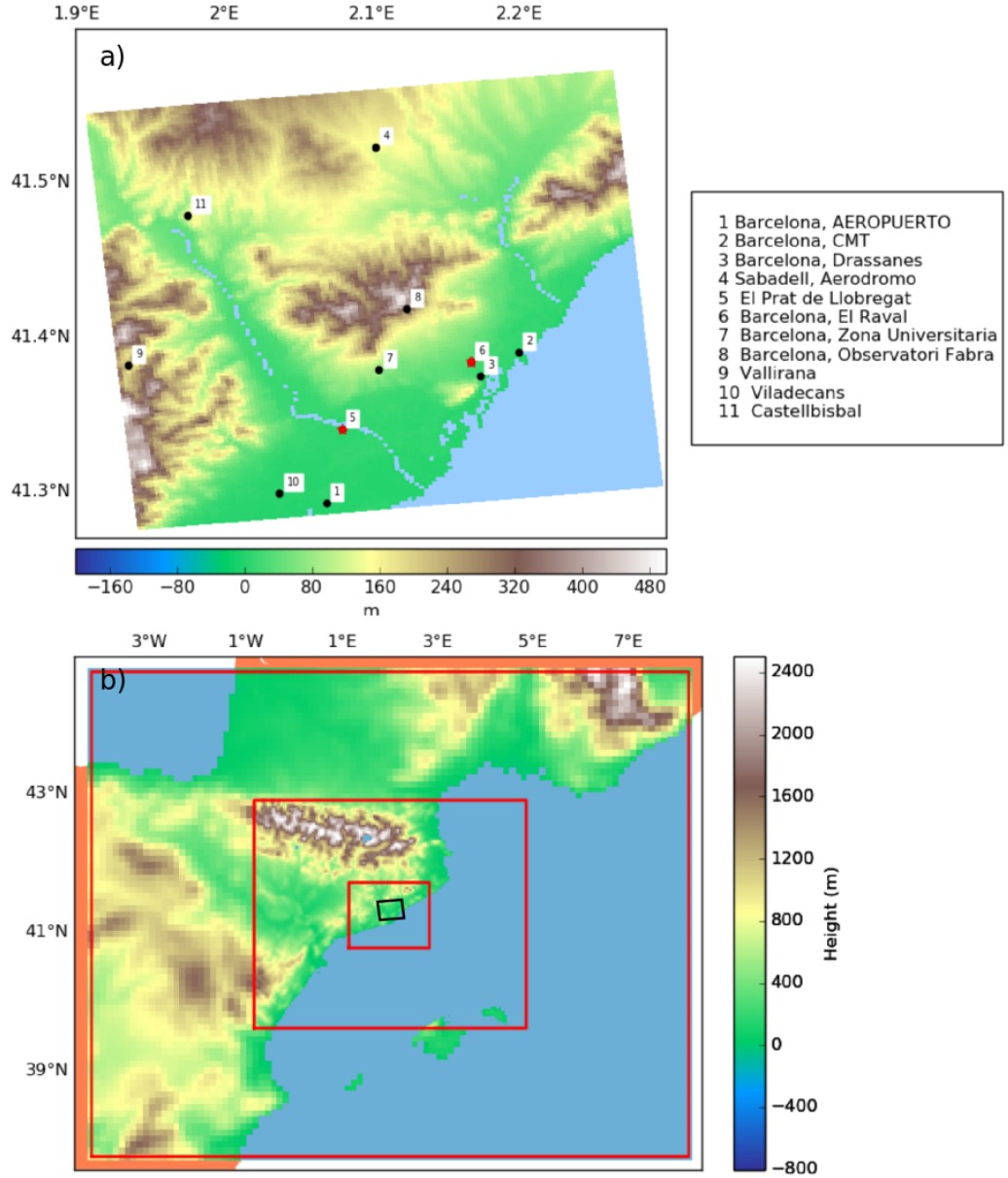

pen classification) is shaped in summer by the local wind breeze regime, whose diurnal evolution exhibits a clockwise rotation from southerlies in the morning to winds blowing roughly parallel to the southwest-northeast shoreline in the late afternoon (Redaño et al., 1991). The main goals of the paper are:

– Evaluation of a UrbClim simulation of the urban climate in the city of Barcelona, driven by reanalysis data at 70 km resolution, against station and satellite data.

– Comparison, both in terms of model skill and computational resources, of the UrbClim simulation against a benchmark simulation performed with a state-of-the art mesoscale model, driven with the same reanalysis data.

– Analysis of the sensitivity of the UrbClim simulation to the boundary conditions, comparing the original run against a simulation driven by a higher resolution forecast dataset (15 km).

## 2 Data and Methodology

### 2.1 Surface stations

As a first approach in the evaluation of the model performance, we have used data from a set of meteorological stations, four of them belonging to the Spanish Meteorological Agency (AEMET) and seven to the Catalan Meteorological Service (SMC). Both data providers carry on a quality control to this data before distributing it. All are well maintained automatic stations, that deliver meteorological data with 10 or 20 min frequency. In the present work, only hourly data were used. The locations of these stations, as well as their names, are displayed in figure 1a, together with the topography.

Station number 5 (El Prat de Llobregat) is chosen to be representative of a rural location near the city. This station is surrounded by cereal fields (figure 2), located 300 m from the Llobregat river and 650 m from the closest urban area. Station number 6 (el Raval) is instead chosen as the reference urban station. This station is located on the roof of a building, at the city center of Barcelona, 8.5 km away from the rural station. Pictures for the locations of both el Raval[2] and el Prat[3] stations are available on the internet site of SMC. These two points are almost the closest possible rural-urban points located at a similar height. The rural station is at 8 m above sea level, while the urban station is at 33 m, which can account for a difference of 0.15-0.25 °C difference in a standard atmospheric profile. The rural station is located in a delta, and therefore the surrounding topography is flat, with no relevant orographic objects between the two stations. Thus, the differences between these two stations are considered to be representative of the UHI effect in the city of Barcelona.

### 2.2 Satellite data

The spatial pattern of the simulations is evaluated through data from the MODerate Resolution Imaging Spectoradiometer (MODIS) of the National Aeronautics and Space Administration (NASA) of the United States. Following previous works (Schwarz et al., 2011; Zhou et al., 2015), MODIS

---

[2]http://www.meteo.cat/observacions/xema/dades?codi=X4
[3]http://www.meteo.cat/observacions/xema/dades?codi=XL

datasets MOD11A2 and MYD11A2 (version 5) were downloaded and processed. These correspond to the Terra and Aqua satellites respectively, and are 8-day aggregations of the daily MOD11A1 and MYD11A1 datasets, using only the clear-sky days. The variable considered is Land Surface Temperature (LST), which is derived from the infrared radiance and emissivity estimated from land cover types. A more detailed description of the algorithms is available in Wan (2008).

The LST data were processed considering only the data flagged as "good quality, not necessary to examine more detailed QA" in the Quality Flag provided with the data, and with no cloudy days during the 8 day period. This introduces a bias to certain meteorological conditions (clear-sky days), which is unavoidable. MODIS and UrbClim LST data were interpolated to a 0.01 deg. regular grid for direct comparison. Finally, only the images with less than 14% missing values were used (this percentage does not include the data over the sea which are always missing). The days (and the times of the day) considered in the averages are the same in the model data and in the observations. The 8-day averages flagged as containing cloudy days in MODIS were masked before computing the averages in both sides. This process left a total of 15 values for most gridpoints (supplementary figure 1) over the whole period.

## 2.3 The UrbClim model

The UrbClim model is designed to simulate the temperature and heat-stress fields at a city scale requiring a minimum amount of computational power, so that it is possible to perform long runs at a resolution of hundreds of metres. A detailed description of the model is available in De Ridder et al. (2015). UrbClim models the lower 3 km of the atmosphere, and consists of a 3-D boundary layer model and land-surface scheme with urban physics. As in mesoscale models, the boundary data is imposed from a lower resolution model, but the boundary conditions scheme somewhat differs. Appart from the variables usually fed to the mesoscale models, UrbClim needs also the radiative fluxes and the precipitación. Instead using a relaxation zone, UrbClim imposes the driving model data in the inflow boundary points, and a "zero gradient" condition, that lets the perturbations flow outside the domain, in the outflow points.

Mesoscale models develop their own variability and structures respect to the lower resolution models driving them, which is called internal variability (Giorgi and Bi, 2000). By design, UrbClim does produce a significantly smaller internal variability than a mesoscale model. This can be considered as a disadvantage, as it does not permit studying the full two-way interaction of the city with all the regional troposphere. But, on the other hand, it is the key for saving computational power. As the UHI is rooted in the surface properties and the heat storage in the ground, using a model like UrbClim is reasonable. This model does work approximately as a wind tunnel, without creating regional structures in the atmospheric flow, so it is possible to nest it directly in much lower resolution models without creating intermediate nests. Nonetheless, this resolution jump can affect the quality of the simulation if the driving model does not accurately reproduce the local climate. Mesoscale

**Figure 2.** Distribution of the land use types used in the UrbClim simulations, which were derived from the CORINE dataset.

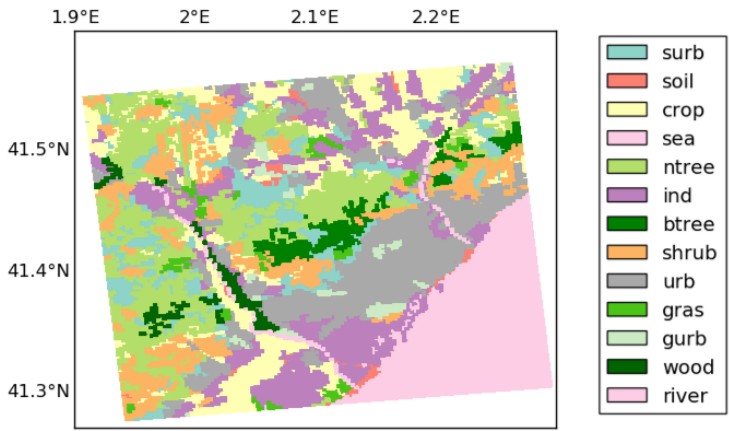

models need these intermediate nests for the inconsistencies between internal variability and driving data not to blow up the numerical stability. This trade-off between the internal variability and the computational efficiency will be key for the interpretation of the results in this study.

The land use data, which are needed to represent the surface properties, are taken from the CORINE dataset[4]). This dataset is publicly available online, and was produced by the European Environmental Agency at a resolution of 100 m. Figure 2 shows the distribution of the land use classes used by UrbClim.

The land-surface scheme is a standard soil-vegetation-atmosphere model based on Ridder and Schayes (1997), which was extended to fully account for the urban canopy. This updated version is described in detail in (De Ridder et al., 2015). The 3D boundary layer model represents a simplified atmosphere by using the conservation equations for the horizontal momentum, potential temperature, specific humidity and mass. The turbulent vertical diffusion is represented following Hong and Pan (1996). The urban physics use a urban slab, together with a parameterization of the inverse Stanton number. This simple approach is justified in De Ridder et al. (2015), because the heat coefficients can be taken from real-world experiments, rather than the scale experiments that the more detailed urban canopy models use to get the transfer coefficients of walls, roofs and roads. In contrast, the Single-Layer Urban Canopy Model (SLUCM) included in WRF represents simple symmetrical street canyons with infinite length (Kusaka et al., 2001). It is important to note that the goal of the present study is not no compare this UCM with the one used by UrbClim, but the whole WRF-SUCM modelling system. Thus, the differences found between WRF and UrbClim are not necessarily related to the different approximations used to represent the Urban Canopy.

---

[4]http://www.eea.europa.eu/publications/COR0-landcover

## 2.4 Experimental setup

**UrbClim**

2.3 The UrbClim simulations cover the five warmest months of year 2011, i.e. from May to September. The domain is represented by a horizontal grid with 121x121 points at a resolution of 250 m, with 19 vertical levels within the 3 lower km of the troposphere (figure 1a). The driving model data is updated every 3 hours. Two simulations have been studied, labeled as UC-ERA and UC-FC. The former is driven by the ERA-Interim reanalyses (Dee et al., 2011), while the latter is driven by the Integrated Forecasting System (IFS) version 37r2 global forecast model of the European Centre for Medium-Range Weather Forecasts (ECMWF). In 2011, this model ran with a spectral resolution of T1279 ($\simeq$15 km), in contrast with the T255 ($\simeq$70 km) of ERA-Interim. Thus, it is able to provide more local details, which can be important given the aforementioned mesoscale-driven weather of Barcelona.

**WRF**

The Weather Research and Forecast model is an open-source, non-hydrostatic limited area model (Skamarock et al., 2008). Thanks to its availability, it has a large community of users. These contribute to the development of WRF, which is leaded by the National Center for Atmospheric Research (NCAR). One particularity of this model is that it has a large amount of parameterization schemes, dynamical options and sub-modules, available to the user to choose among them. These options are set up in a namelist file that must be edited for each simulation.

In the present study, we used the version 3.6.1 of WRF, which was configured to run in three nested domains (figure 1b), with 40 vertical levels and horizontal resolutions of 10, 3.3 and 1.1 km. The 250 m of UrbClim were not reached because the computational cost was not affordable. However, the simulations were carefully configured to make them comparable with UrbClim: they were nested in the same dataset (ERA-Interim) and used the same land use (CORINE). WRF land use is taken by default from the United States Geological Survey (USGS) dataset. Thus, the CORINE land classes were mapped to the USGS 33 classes following table 7.1 of Chrysoulakis et al. (2014). The resulting land use class map is shown in figure 3

Despite being nested in a reanalysis, the regional models tend to generate their own internal variability. While this is necessary to the RCM to add value, it is not convenient to let the model to drift too much from the reanalyses, as these incorporate observations and are accurate descriptions of past atmospheric states. There are two approaches to solve this: using nudging, or restarting the model frequently. In this case, based on the experience of previous studies (Menendez et al., 2014; García-Díez et al., 2015), daily 36 hours simulations have been carried out and concatenated, leaving 12 hours as spin-up. These simulations cover the same time span as UrbClim, from May to Septem-

**Figure 3.** Distribution of the land use types used in the WRF simulation. They were mapped from the CORINE dataset to the USGS classes.

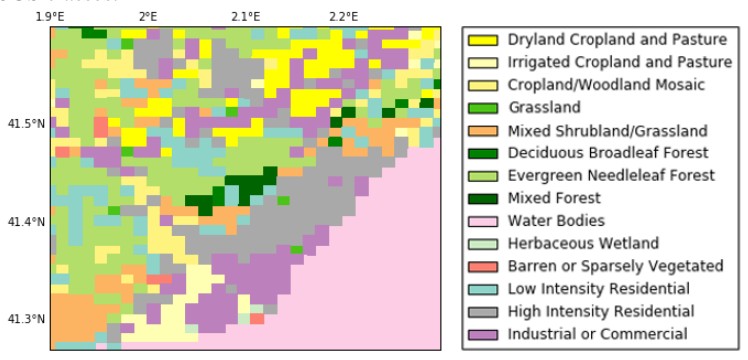

**Table 1.** Scores of daily mean 2 metre temperature for the UC-ERA, UC-FC and WRF simulations and the 11 stations depicted in figure 1. The scores are: Mean bias (model - observed), Root Mean Squared Error (RMSE), and variance ratio (i.e. the variance of the model divided by the variance of the station).

| Station | Bias (°C) | | | RMSE (°C) | | | Variance ratio | | |
|---|---|---|---|---|---|---|---|---|---|
| | UC-ERA | UC-FC | WRF | UC-ERA | UC-FC | WRF | UC-ERA | UC-FC | WRF |
| 1 | 1.4 | 0.1 | -0.5 | 2.0 | 0.9 | 0.9 | 1.1 | 0.8 | 0.7 |
| 2 | 2.0 | 0.7 | -0.8 | 1.9 | 0.8 | 0.7 | 1.6 | 1.1 | 0.9 |
| 3 | 1.6 | 0.3 | -0.8 | 1.9 | 0.8 | 0.7 | 1.5 | 1.1 | 0.9 |
| 4 | 2.1 | 1.1 | 1.1 | 2.1 | 1.2 | 1.1 | 1.4 | 1.0 | 1.0 |
| 5 | 1.1 | 0.2 | -0.3 | 1.7 | 0.6 | 0.7 | 1.3 | 1.1 | 0.9 |
| 6 | 1.2 | 0.1 | -1.0 | 1.8 | 0.7 | 0.6 | 1.5 | 1.1 | 0.9 |
| 7 | 1.8 | 0.5 | 0.6 | 1.8 | 0.7 | 0.8 | 1.4 | 1.0 | 0.9 |
| 8 | 0.6 | -0.7 | 0.6 | 1.8 | 1.1 | 0.9 | 1.1 | 0.8 | 0.8 |
| 9 | 0.5 | -0.3 | -0.2 | 1.6 | 0.7 | 0.7 | 1.1 | 0.9 | 0.9 |
| 10 | 1.2 | -0.2 | 0.3 | 1.7 | 0.6 | 0.7 | 1.4 | 1.0 | 0.9 |
| 11 | 0.3 | -0.5 | -0.0 | 1.7 | 0.8 | 0.8 | 1.2 | 0.9 | 0.9 |

ber 2011. Thus, 153 individual simulations have been carried out. To handle them, the WRF4G
framework (Fernández-Quiruelas et al., 2015) has been used.

## 3 Results

### 3.1 Time series

Table 1 shows scores of daily mean 2 m temperature for the UC-ERA, UC-FC and WRF simulations and the 11 stations. The largest errors are found in UC-ERA, which generally overestimates daily temperatures by up to +2°C at some stations. This overestimation is associated with the misrepresentation of the sea breeze, which has larger effect on maximum temperatures. This is discussed later

**Figure 4.** Average daily temperature cycle in the urban (left) and rural (middle) stations. The difference urban minus rural is shown in the panel on the right. The "UC-ERA", "UC-FC" and "WRF" legend codes are defined in section 2, while "OBS" represents the observation.

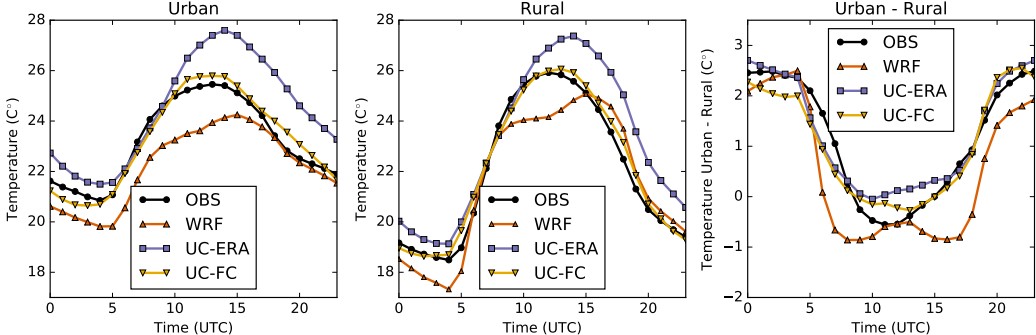

in the manuscript. UC-ERA also overestimates the day-to-day variability, having higher Root Mean Square Error (RMSE) than the other runs. Instead, UC-FC and WRF show similar, smaller scores, which indicate the good performance of these simulations.

Figure 4 shows the average daily cycles for the urban and rural stations, as well as their difference. The average magnitude of the UHI during the night is found to be 2.5°C, which is large enough to have direct impacts on human health during heat wave episodes (Ye et al., 2012). During daytime hours, the UHI is found to decrease down to -0.5°C. Note that this is in very close agreement with the values derived from observational data in Moreno-García (1994), despite it used two different

reference points. The measurement of the UHI with only two points has some limitations, as it may be sensitive to very local features such as the land use in the vicinity of the stations. However, the representativeness of these points has been carefully checked with high resolution satellite images. In addition, the agreement with previous studies increases our confidence in the results here presented.

       UC-ERA tends to overestimate temperatures at both stations after 10 UTC and particularly during

daytime hours, but errors in both stations cancel each other, and therefore the UHI magnitude is generally well represented with biases smaller than 0.5°C. The UHI average daily cycle is similar in UC-FC and UC-ERA, but UC-FC does not show any warm bias, and accurately reproduces the observed temperatures of the individual stations.

       In the case of WRF, we initially considered the nearest gridpoint to the rural and urban stations,

and biases in the three panels were found to be clearly larger than those in UrbClim (not shown). This problem was found to be related to the land use of the gridpoints, which were not representative of the land use of the stations. Indeed, the gridpoint representing the rural station was found to be classified as urban in the land cover map used by WRF. In order to address this problem, we considered a more representative, adjacent gridpoint to represent the rural station, which is used

throughout the paper (see supplementary figures 1 and 2 for the details). Results show that biases in WRF for the individual stations are large and negative throughout the day except in the evening, but

**Figure 5.** Same as figure 4, but for wind speed.

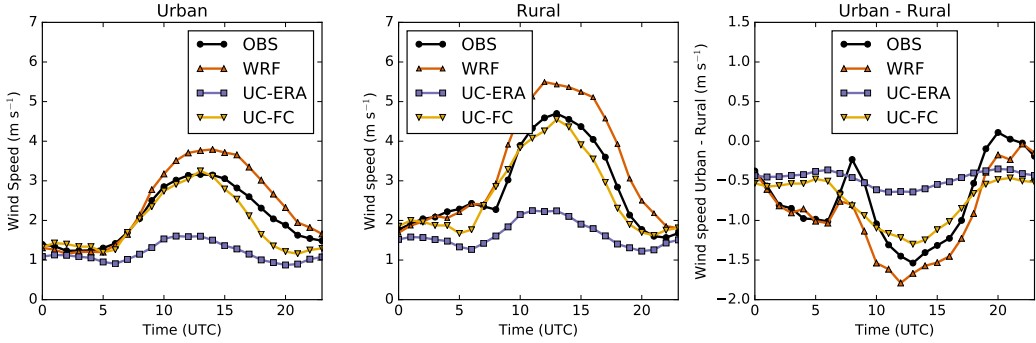

comparable in magnitude to those in UC-ERA. In addition, although the UHI at noon is correctly reproduced by WRF, its bias is clearly larger at 7 (-1.5°C) and 17 UTC (-1°C).

Regarding the wind speed (figure 5), the intensity of the sea breeze is clearly underestimated in
UC-ERA, with a bias of up to -2.5 ms$^{-1}$ at noon in the rural station. This problem is likely to be related with the coarse resolution of the ERA-Interim driving dataset, which is not able to resolve the sharp daytime, thermally-driven pressure gradient between the continent and the sea. The lack of sea breeze in turn explains the nearly constant daily cycle of the rural minus urban difference in wind speed in UC-ERA. The wind regime is clearly better reproduced in the other simulations.
UC-FC accurately reproduces the daily wind cycle in both the urban and rural stations, while WRF overestimates the wind speed by up to 1 ms$^{-1}$ during daytime hours. Regarding the urban minus rural difference, WRF is the model that better captures the hourly evolution of the wind speed. UC-FC correctly simulates the overall magnitude of the difference, but it is not able to reproduce the secondary minima and maxima at 6 and 8 UTC.

It is interesting to highlight the day-to-day variability of the observed and simulated times series for the month of May (figure 6). The whole period is not shown for clarity, but the same conclusions are applicable for the other months. The daily evolution of the UHI is well represented in UC-FC and WRF, while biases of the order of up to 4°C at noon are found in UC-ERA for some specific days. However, the largest Mean Absolute Error (MAE) is found in WRF (1.11°C), due to the systematic
underestimation of the UHI during daytime hours (figure 4). This underestimation is small, albeit persistent. The best MAE score is found in UC-FC (0.80°C), which shows regular skill with almost no large errors in specific days.

From these results, it is clear that the performance of UrbClim is largely improved as a result of the higher resolution of the ECMWF forecast model compared to ERA-Interim (15 km vs. 70 km).
It is however unclear how to understand the comparison of WRF with UrbClim. On one hand, UC-ERA and WRF, which are both nested in the same reanalyses, display comparable scores regarding the magnitude of the UHI, althoguh WRF better represents the wind speed and some spatio-temporal features of temperature. However, WRF performs a full dynamical downscaling down to a resolution

**Figure 6.** Hourly time series of the 2 meter temperature difference between the urban and the rural locations for May 2011, for WRF (top), UC-ERA (middle) and UF-FC (bottom).

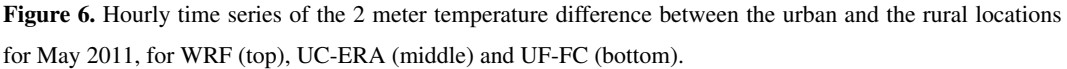

of 1.1 km, and therefore, in principle it should be able to achieve an accuracy similar to UC-FC.
But, as we have shown, UC-FC exhibits better scores than WRF. Given the large number of factors involved, it is difficult to find an explanation to this result in physical terms. In general, WRF is more biased than UC-FC (figures 4, 5 and 6) and than the ECMWF forecast itself (not shown). It tends to underestimate temperatures and to overestimate the wind speed during the day. As WRF is very customizable, it could be possible, in principle, to find a configuration that removes these biases.
However, WRF biases in wind speed are found to be difficult to correct, and research is yet ongoing in this line (García-Díez et al., 2015; Lorente-Plazas et al., 2016).

**Figure 7.** Daily minimum temperature averaged over the period May-September 2011 in a) UC-ERA, b) UC-FC and c) WRF.

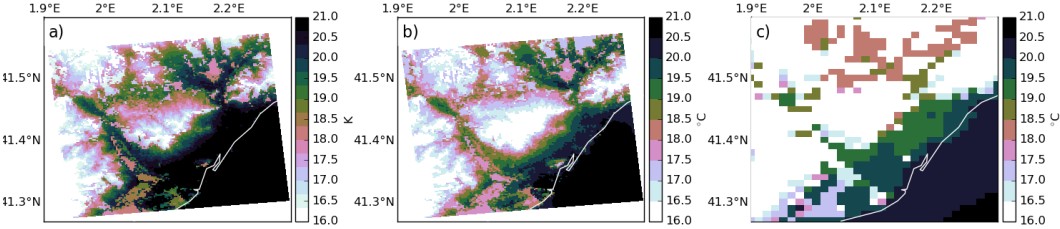

**Figure 8.** Land surface temperature averaged during nighttime hours over the period May-September 2011 in MODIS (left), UC-ERA (center) and WRF (right).

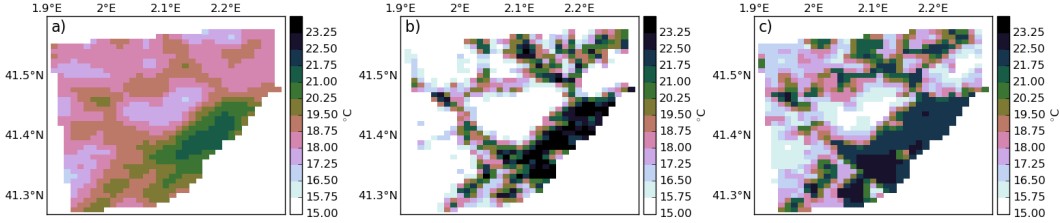

## 3.2   Spatial pattern

The evaluation of the spatial variability simulated by the urban climate model is a challenging issue due to the lack of reliable, high-resolution observations. Figure 7 shows the average daily minimum

temperatures for UC-ERA, UC-FC and WRF, for the 5 months considered. Although both models are able to resolve the main features of the UHI of Barcelona, the surrounding cities and the airport, the UrbClim run at a resolution of 250 m provides much more detailed information, e.g. a clear representation of the hill to the southwest of the city centre. UC-ERA is generally warmer than UC-FC, due to the mis-representation of winds, but the spatial patterns are very similar. In addition, the

temperature difference is similar in the urban and rural sites. Unfortunately, the scarcity of surface observations did not allow us to evaluate the spatial patterns at the screen level, and therefore we evaluated the spatial variability of the model by analysing the MODIS satellite LST, as described in section 2.2.

During the night, both UC-ERA and WRF have been found to overestimate LST over urban ar-

eas, and therefore also the LST UHI (figure 8). This is surprising, given the small error found in the evaluation of the screen level UHI. Other studies (Zhou et al., 2015) found small errors when comparing MODIS and UrbClim LST in and around the city of London. As mentioned in the introduction, measuring LST over urbanised areas is challenging, due to the uncertainties associated with the measurement of both the radiation and the emissivity. By comparing modeled (panels b and

c) and observed (panel a) maps in figure 8, it can be seen that the bias outside the urban areas is found to be relatively small in WRF, and slightly negative in UC-ERA, while the spatial patterns are

reasonably similar between the models and the satellite data. Determination of emissivity over urban areas is notoriously difficult and subject to large uncertainties, which could explain at least part of model deviation of LST. The spatial Pearson correlations between the observed and simulated fields are 0.77±0.025 for UC-ERA and 0.70±0.03 for WRF, where the confidence bounds were computed with bootstrapping (1000 samples). Thus, the spatial correlation in UC-ERA is higher, and the difference is statistically significant. However, taking into account the above-mentioned differences in the bias, we conclude that the performance of the spatial pattern in WRF and UC-ERA are comparable.

It is worth mentioning that the MODIS LST appears to have an effective resolution coarser than 1 km, given that the spatial patterns are smooth and do not resolve many detailed features. Finally, as mentioned in the introduction, LST and Surface Air Temperature (SAT) UHIs are not equivalent, and are driven by different phenomena. Thus, it is also possible that models that reproduce the SAT UHI correctly generate at the same time a biased LST UHI.

### 3.3   Computational resources

In this section, the computational resources required by UrbClim and WRF are compared. The comparison is not fully trivial because UrbClim does not currently support running in parallel, which can be seen as an important drawback. However, UrbClim does not require a long spin-up, and therefore the simulations can be parallelised just by splitting the time period in subperiods and run the corresponding simulations simultaneously in different machines or nodes.

For a direct comparison, both models were run in the local cluster of the Institut Català de Ciències del Clima (IC3), while the main UrbClim runs used in the paper were carried out in the VITO cluster. The IC3 cluster is made of 48 homogeneous server blades, having each of them two "quad core" processors, 48GB of memory, 146GB of disk space and fast network interconnect (Infiniband). The blade model is Sun Blade X6270 (see http://www.oracle.com/us/products/servers-storage/servers/blades/sun-blade-x6270-m2-ds-080923.pdf for a full description) equipped with Xeon (Nehalem) X5570 processors. Results are summarized in table 2. With these settings, a WRF simulation of 36 hours took 2.5 hours to finish (using an average of 10 simulations), including the preprocess carried out with the WRF preprocessor (WPS). This preprocess was run in serial, in 1 core, while WRF was run in 16 cores, this is, two blades. Thus, the total serial equivalent wall-time was 40 hours, assuming perfect scaling (the real value will be somewhat below). WRF was compiled using the Intel fortran compiler version 14.0.1 with the Intel MPI Library for Linux OS, Version 4.1 Update 3.

Regarding UrbClim, for this test, it has been compiled with the same compiler and run in the same cluster. A 36-hours simulation with UrbClim took 0.3 hours to finish (average of 10 simulations) running in one core. Thus, UrbClim running at 250 m resolution is found to be 133 times faster than WRF at 1 km resolution. This enables downscaling large climate change ensembles for a big collection of cities.

Note that the UrbClim speed is not only explained by the smaller number of gridpoints (table 2), but especially because of the simplicity of the dynamical core, and the smaller number of parameterizations, compared to WRF.

## 4 Conclusions

In the present work, we have evaluated the performance of a boundary-layer urban climate model (UrbClim) for the warm season in the city of Barcelona. We were particularly interested in the study of the UHI effect, given that it represents a major source of health problems in summer for vulnerable people living in urban environments (e.g. heat stress, temperature-related mortality, pollution, vector-borne diseases). We have compared this model (UC-ERA) with the output of a regional climate model (WRF), and analysed the effect of the model resolution in the driving simulation (UC-ERA vs. UC-FC). All these simulation have been evaluated against observations from meteorological stations and satellite data (MODIS), in order to analyse the temporal and spatial variability of the UHI effect, respectively.

The main conclusions of our work can be summarised as follows:

- The average UHI in the city of Barcelona during the warm season (May-September) reaches 2.5°C at night. This is relevant for the study of climate impacts, given that it increases the stress to the vulnerable population and for the health care systems under extreme conditions.

- UrbClim correctly reproduces the UHI of Barcelona when it is nested to the coarse dataset of ERA-Interim, while it suffers from a general warm bias. When it is nested to a higher resolution model (ECMWF IFS), UrbClim additionally reproduces well the temperature evolution of the individual rural and urban stations used for the calculation of the UHI.

- WRF is less biased than the UrbClim run nested in ERA-Interim, and both runs show comparable skill on reproducing the UHI.

- The spatial pattern of LST is similar in UrbClim and WRF, even though significant biases are found in both models when they are evaluated against MODIS data.

In conclusion, UrbClim has been found to be well suited for the numerical description of the UHI of Barcelona, providing an accurate description of the temperature field. The choice between UrbClim and WRF for the simulation of the urban environment largely depends on the type of variable and process that is to be analysed. WRF has the advantage of providing a more detailed and complete description of atmospheric winds and rainfall, which is required in some applications (e.g. pollutant dispersion, urban effect in rainfall). On the other hand, UrbClim has been proven to be as accurate as WRF on reproducing the UHI of Barcelona during the warm season, and several orders of magnitude faster. This opens the door to the performance of multi-decadal simulations of urban

**Table 2.** Summary of the results of the benchmarking. The number of gridpoints represents the total number of gridpoints in the model domain, which is 121*121*19 in UrbClim, and 100*80*40 + 136*112*40 + 121*97*40 in case of WRF (taking into account the three nested domains).

| Model | Number of gridpoints | Horizontal resolution | Time step | Wall-time for 36h |
|---|---|---|---|---|
| UrbClim | 278,179 | 0.25 km | adaptative[5] | 0.3 h |
| WRF | 1,398,760 | 1.1 km (10x3.3x1.1) | 60 s | 40 h |

heat stress in a large number of cities at a reasonable computational cost, using multi-scenario and multi-GCM ensembles to account for uncertainty, and testing for urban adaptation scenarios.

We found that, in cities affected by strong mesoscale flows (e.g. sea breeze) such as Barcelona, it must be taken into account that UrbClim will be subject to inaccuracies caused by the mis-representation of the wind, in case that it is nested in a low resolution model.

Note that this is a specific problem of Barcelona, given that it has not been found in other European cities where UrbClim driven by ERA-Interim has been successfully tested (De Ridder et al., 2015; Lauwaet et al., 2016; Zhou et al., 2015). From these results, it is reasonable to infer that the skill of UrbClim, and probably of other similar urban boundary layer models, is constrained by the performance of the driving model, and particularly for variables that are important for the UHI, this

is, wind speed and cloudiness.

**Code and data availability**

The Urbclim source code is not publicly available. In order to access it, a specific agreement needs to be signed with VITO. Please contact koen.deridder@vito.be for more details. The WRF model is an open source model, and its code is freely available upon registration in http://www2.mmm.ucar.edu/

wrf/users/download/get_source.html. Weather station data from the Catalan and Spanish meteorological agencies is available for research purposes upon request in dades@meteo.cat and https://sede.aemet.gob.es/AEMET/es/GestionPeticiones/home respectively. MODIS data were downloaded from the "Reverb" NASA tool http://reverb.echo.nasa.gov, where it is freely available upon registration. The CORINE land cover is available in the EEA website http://www.eea.europa.eu/publications/

COR0-landcover free of charge for both commercial and non-commercial purposes.

*Acknowledgements.* The work described in this paper has received funding from the European Community's 7th Framework Programme under Grant Agreements No. 308299 (NACLIM) and 308291 (EUPORIAS). J. Ballester gratefully acknowledges funding from the European Commission through a Marie Curie International Outgoing Fellowship (project MEMENTO from the FP7-PEOPLE-2011-IOF call), and from the European

---

[5]20 s for the soil scheme and adaptative for the atmosphere, using the Courant-Friedrichs-Lévy stability criterion.

Commission and the Catalan Government through a Marie Curie - Beatriu de Pinós Fellowship (project 00068 from the BP-DGR-2014-B call).

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
