# Peer review of "Advantages of using a fast urban boundary layer model as compared to a full mesoscale model to simulate the urban heat island of Barcelona"

_Geoscientific Model Development, 2016_

## Referee Comment (RC1) · Anonymous Referee #1 · 3 Mar 2016

"general comments"

This paper adresses interesting results on the scope of the presented topic. The methodology is clearly described and the results support the reached conclusions.

From my point of view, the manuscript has the quality required to be published after some minor technical corrections. However, I missed some more details on the analysis of results showed in figure 6 (see "specific comments").

"specific comments"

Line 97-98: I disagree with this argument: "the surrounding topography is flat, which does not favour temperature inversions during night time hours". In fact, flat terrain

favours the development of a nocturnal stable layer. However, since inversions are more related on atmospheric stability, I suggest to change this argument. Perhaps you just can comment that there are no significant orographic objects between the two stations.

Line 239-240: Do you think that the LST overestimation of both UrbClim and WRF can be related to the fact that the cloudy nights are not considered after filtering the satellite images as it is described in chapter 2.2? In other words, to compare both models against satellite, the same days have been considered to compute the LST averaged during nightime hours for the warm season (figure 6)?

Line 287: I think it is important to emphasize that the first conclusion is obtained after analysing observational data and valid only for the warm season.

"technical corrections"

Line 93: style. Substitute "...at the city centre of the city..." for "...at the city center of Barcelona..."

Line 146: typing error. Substitute "...important give the..." for "...important given the..."

Line 154: typing error. Substitute "...nested domains 1b..." for "...nested domains (figure 1b)..."

Line 242: typing error. Remove "advectionAs"
* * *

---

## Referee Comment (RC2) · Anonymous Referee #2 · 9 Mar 2016

This paper is aimed at showing the advantages of an off-line urban canopy model with respect to a regional climate model coupled to an urban canopy model for urban heat island studies. It is undeniable that there is benefit in the use of application-specific models such as UrbClim, particularly in terms of computational costs. In such circumstances, even if they both show similar results, the computational costs associated with the regional climate model make the use of UrbClim very attractive. However, in order to justify the use of a standalone urban model, the authors compare completely different tools (i.e. a more like-to-like comparison would be to show benefits of UrbClim over the Single-Layer Urban Canopy Model, both running offline with the same boundary conditions) and do not fully acknowledge (only at the very end of the paper)

that a regional climate model coupled with an urban model incorporate features that an offline model cannot (e.g. two-way interaction with surrounding circulation such as sea-breeze). But what is more important, the comparison is to a large extent unfair because the authors claim a better representation of local temperatures when the Urb-Clim is driven by ERA-Forecast, but do not test the regional climate model running with the same large/mesoscale information. Finally, the authors do not provide any explanation of what processes are better represented in UrbClim that make it perform better than the regional climate model?. In my opinion, the starting point is not correctly posed and the authors do not adequately support their conclusions with a rigorous analysis. I agree with the authors that for this particular application, UrbClim might present advantages over a Regional Climate Model coupled with an urban model, but I don't think the authors provided enough evidence for that.

In my opinion, the authors could make additional experiments, perform a like-to-like and more in-depth comparison, with possible reasons as to why UrbClim outperforms the RCM. In that case, they should also mention that RCMs are a tool design to conduct atmospheric research and therefore have a wider range of applications, and this is the reason why they are selected over offline and faster models.

As it is, I am unsure the paper makes a scientific or model development contribution worth publishing. Perhaps including the additional analyses suggested above could lead to a paper that is adding to the current knowledge. In addition to some general comments, I have also suggested some specific and technical comments aimed at improving a future version of the manuscript.

Therefore, I would not recommend this paper for its publication at Geoscientific Model Development.

General comments

1.- The authors mention internal variability in multiple occasions, but it is unclear from their discussion what they mean by internal variability. It is also unclear why internal

variability is regarded as intrinsically negative (L161-165). I understand that the authors preference is to avoid model departures from the boundary conditions in terms of the large scale (or mesoscale) conditions and I agree that in that sense, UrbClim does not generate any "internal variability" (L178), but in a broader sense, it is required that models produce some internal variability so they produce results the are different from the boundary conditions (added information). In any case, the authors need to be more explicit about what they mean by "internal variability". The authors also mention internal variability to justify the resolution jump between the boundary conditions and the UrbClim resolution (L121-123). I don't quite understand the sentence and why it is acceptable to have such a difference between resolutions. UrbClim is effectively downscaling a single grid point of ERA-Interim and therefore forced everywhere with the same conditions. This is something that compromises the representativeness of the results.

2.- What does UrbClim do better than WRF-SLUCM? It is necessary that the authors provide a better description of the models and a reasoning of which processes UrbClim might be improving. It is necessary to describe how are the city characteristics seen by the models, what are the differences in the two models (e.g. extension, density, types of building, vegetation cover. . .) Are they different in each of the models? What are the differences how UrbClim deals with urban and rural areas? And WRF-SLUCM?

3.- The extension of UrbClim domain is roughly 25 km by 25 km. UrbClim is essentially an offline model driven by 1-grid point from ERA-Interim and 4 grid points in ERA-FC. It is obviously much faster than the RCM (considering the number of grid points, including the vertical, in each model provides an idea of the considerably lower computational requirements of UrbClim. This is a very important feature, but it should be considered that the area simulated is much smaller and the applications more limited.

4.- The comparison is unfair in many different ways. UC-FC and WRF cannot be compared in any way (They are forced by different mesoscale conditions). In the abstract, the author anticipate a better performance of UC-FC, but in my opinion this result does

not prove that UC outperforms the RCM. The authors are evaluating at the same time the performance of ERA-Interim and ERA-FC, and UrbClim and WRF. This comparison would only be acceptable if WRF is driven by ERA-FC too. I would suggest the authors consider other experiments to adequately assess the performance of the models. In addition to WRF driven by ERA-FC, UrbClim could be driven by WRF outputs and test if it improves WRF estimates.

6.- The question that arises after reading the manuscript is, why not using statistical downscaling, which is even faster, allows for large ensembles and will reproduce present climate results much better as the are purposely calibrated. Why would an user opt for UrbClim over statistical downscaling? I would guess the answer is the spatial coverage of UrbClim.

Specific comments

L44 Regional climate models have multiple applications, not just downscaling climate projections. Perhaps: "RCMs are limited area models used to downscale climate change projections from coarse resolution Global Circulation Models as well as other applications."

L53 Please specify which of the Urban Canopy options if the authors want to keep the following sentence. "This parameterisation...". Otherwise, specify only in the methods. Also, if the authors refer to Single-Layer Urban Canopy Model, it was developed by Kusaka et al. (2001). Chen et al. (2011) describe the implementation in WRF. I also miss some references to work done with WRF/SLUCM for future projections (e.g. Georgescu et al. 2013, Argueso et al. 2015, Kusaka et al. 2012).

L64 Add ", especially" after urban pollution. Urban pollution exacerbates sensitivity to adverse conditions is for all population segments.

L70 (FIGURE 1) I would be necessary to have some sort of reference of the urban extension in Figure 1. For those of us not familiar with the region it is difficult to locate

the city, and urban areas in general, in the map. It is a major feature of this study and should be shown.

L93 How representative is the temperature in the roof of a building of the temperature at levels that matter for population? How comparable is that to the temperatures provided by both models at 2 m .

L97-99 I disagree with this statement. Flat areas are indeed characteristics of temperature inversions, particularly near the coast. Please revise.

L112-113 In the estimate of missing values, did the authors considered all land points within the UrbClim domain? Why 14%?

L116 What to the authors mean by "main features of the urban climate"? Are there hydrological variables (e.g. precipitation, evaporation).

L132. In which ways? If the extension is not described, this sentence can be removed.

L164-165 It is not clearly described how this configuration deals with soil variables. Are they obtained from ERA-Interim at every reinitialisation? If so, NOAH LSM is constantly trying to balance the information from ERA-Interim LSM and they are not necessarily compatible (the even don't share the same layers). A similar question arises for UrbClim in terms of how it sees soil temperature and moisture.

Table 1 How is the variance ratio calculated? What does it represent?

L170-174 Do these values in UC-ERA ultimately depend on information provided by a single grid point from ERA-Interim? The resolution of ERA-Interim is equivalent to ∼70 km, while the extension of the UrbClim domain is ∼25 km by 25 km.

L179-180. It is not correct to say that the "results can be interpreted as a comparison between Urban Canopy+PBL models driven by" different boundary conditions. In WRF there is a two-way interaction, the results from Urban Canopy+PBL are fed back into the dynamical core and influences local circulation (and potentially larger scale circulation).

L181-182 is especially concerning since the authors disregard that this comparison is mixing multiple things, the mesoscale information to begin with. I agree that at these scales (15km to 1km) and for variables such as temperature, there is little benefit in increasing resolution, but this cannot be inferred from the authors results.

L197-198 Is this a merit of FC over ERA-I rather than UrbClim?

L200-205 It is obvious that if the model was assigning urban land use to the rural location, then for the purpose of this study they are not comparable by any means. I do not think this needs to be mention in the text. Instead, I would say that the closest grid point with a particular land cover (and specify the land cover) was selected to compare with the rural station, and perhaps say the distance between the grid point and the station.

L 206-207 I would say that the bias occurs throughout the day except in the evening (16H-00H)

L 210 This is not surprising. As the authors suggest, at 70 km the intensity of the sea breeze is often underestimated. But also, only information from 1 ERA-Interim grid point is provided to UrbClim.

Figure 3. A physical explanation of why sea breeze is weaker in urban points would be desirable. It cannot simply be increased drag because the urbanised areas along the coast act as a barrier for the rural station too. Furthermore, the urban station is located in a roof, so depending on the surrounding buildings and the height of the station, the drag could be negligible for that station. In this case, the comparison tell us about the boundary conditions, not necessarily the UrbClim. It could be well the case that UrbClim is doing a fantastic job in both cases, but ERA-FC information is more accurate (or more comparable to those scales).

L221-228 It is unclear what the authors want to illustrate with figure 4. What is the contribution to the paper? Isn't this day-to-day variability highly influence by the boundary

conditions?

L 230-238 (Description of figure 5) This is basically qualitative. It does offer finer detail because it has finer resolution, whether this is correct remains unclear, even after comparison with relatively low resolution MODIS (Figure 6), where a more quantitative measure is provided.

L 239-240 (and onwards) Among the multiple descriptions of UHI, two are widely used. The skin temperature (or surface temperature) UHI and the screen level temperature UHI. Although linked, they involved completely different processes. Indeed, skin temperature UHI is generally positive at day and night, but the screen level temperature UHI is often negative during the daytime. The authors should capture this in their discussion.

L 241 I understand that results from Zhou et al. 2015 contrast with the authors finding rather than agreeing.

L249-250 If the authors provide these confidence bounds, both values are exactly the same (not just statistically insignificant). Not only the confidence bounds overlap but the both include the estimate from the other model.

L 300 But UrbClim does not provide rainfall at all, or does it?

Technical corrections

L34-36 Please revise sentence, how it links to the previous one. Please revise use of commas. L37 Replace "defer" with "differ" L38-39 Please revise use of commas. L41 The end of this sentence is unclear. Please rewrite. (The last half of this paragraph needs to be clarified) L59 Remove "now" (?) L93 Replace "Km" with "km" L152 Remove comma after sub-modules. Missing verb before available? L 242 Please remove "advection"

---

## Author Comment (AC1) · 30 Mar 2016

The authors would like to thank the referee for this thorough review. Despite being negative about the paper, we believe that the review is useful, as it raises a number of valid points and criticism that will improve a future version of the manuscript once they are taken into account. As we explain below, we believe that the decision of rejecting the manuscript is not justified enough by the reviewer's comments. If the editor approves it, we will take into account all the reviewer's comments in order to proceed with the rewriting of several paragraphs of the manuscript and a more in-depth interpretation of the results. In this message we will answer the main comments of the reviewer. A more complete answer corresponding to the modifications introduced will be provided after the discussion is closed.

*This paper is aimed at showing the advantages of an off-line urban canopy model with respect to a regional climate model coupled to an urban canopy model for urban heat island studies.*

Note that UrbClim is not exactly an offline urban canopy model (UCM). Apart from an UCM, UrbClim has a planetary boundary layer (PBL) parameterization too, and does carry out a simplified simulation of the lower part of the atmosphere (see section 2.3 of the manuscript and De Ridder et al. 2015). This is an important point that perhaps we did not stress enough in the manuscript, but that we will explain in more detail in the new version.

De Ridder, Koen, Dirk Lauwaet, and Bino Maiheu. 2015. "UrbClim – A Fast Urban Boundary Layer Climate Model." *Urban Climate* 12 (June): 21–48. doi:10.1016/j.uclim.2015.01.001.

*It is undeniable that there is benefit in the use of application-specific models such as UrbClim, particularly in terms of computational costs. In such circumstances, even if they both show similar results, the computational costs associated with the regional climate model make the use of UrbClim very attractive.*

The computational cost is not just an added value, but a critical factor, because it allows the performance of multi-decadal simulations in large ensembles of cities at sub-kilometer resolutions. The cost of doing this with a Regional Climate model (RCM) is not possible to afford. Note that the computational cost is also the very reason of the existence of RCMs.

*However, in order to justify the use of a standalone urban model, the authors compare completely different tools (i.e. a more like-to-like comparison would be to show benefits of Urb-Clim over the Single-Layer Urban Canopy Model, both running offline with the same boundary conditions) and do not fully acknowledge (only at the very end of the paper) that a regional climate model coupled with an urban model incorporate features that an offline model cannot (e.g. two-way interaction with surrounding circulation such as sea-breeze).*

As we pointed out, UrbClim is not exactly an offline UCM. The paper aims to compare UrbClim with a widely used RCM such as WRF, and not to compare UrbClim with the UCM of WRF. The scope of the paper is applied and pragmatic, in the sense that it is not trying to find the best theoretical approach to model the Urban Heat Island (UHI). We try to show that in this case it is not necessary to run the primitive equations in the whole troposphere, together with the parameterizations of radiation, cumulus... etc., as WRF does, in order to reproduce a realistic UHI.

*But what is more important, the comparison is to a large extent unfair because the authors claim a better representation of local temperatures when the Urb-Clim is driven by ERA-Forecast, but do not test the regional climate model running with the same large/mesoscale information.*

This seems to be a major concern for the reviewer. A fair comparison between two completely

different approaches is not fully possible. A regional model like WRF is able to generate his own mesoscale variability so, in principle, it should be able to compete with the UrbClim run nested in ECMWF forecast model, as the large-scale variability is very similar in the two global models used.

In fact, it could be argued that WRF has an advantage here, as its UCM and PBL are running with forcing fields at a 15 times better resolution than those used for UC-FC. These kind of issues make the fair comparison between the two approaches complicated (though the comparison is certainly not fair for the WRF UCM alone, but as mentioned, this is not the goal of the paper). See the following paragraph in the manuscript (L 178-185):

"As UrbClim is hardly able to generate internal variability, these results can be interpreted as a comparison between Urban Canopy+PBL models driven by ERA-Interim (70 km), ECMWF forecast (16 km) and WRF (1 km). Thus, differences in the results show the added value of the higher resolution in the ECMWF forecast model and WRF. However, note that the extra resolution of WRF (about 15 times higher than ECMWF) is not clearly improving the results. This is consistent with previous studies suggesting diminishing returns for added value in this resolution ranges (García-Díez et al., 2015)."

The problem here may be that WRF is more biased than the ECMWF forecast model, which is stunningly  precise. So, that is why we say in the conclusions (L310-312): "From these results, it is reasonable to infer that the skill of UrbClim, and probably of other similar urban climate models, is constrained by the performance of the driving model, and particularly for variables that are important for the UHI, this is, wind speed and cloudiness". This holds also for WRF-UCM. If driven by ECMWF forecast model, WRF-UCM may be able to perform as well as UC-FC. But testing this is out of the scope of the paper.

There may be other RCMs showing better results but, as WRF is currently widely used in this kind of studies about the UHI effect, we think that the results of the paper are relevant. On the other hand, results show that relatively high resolution (16 km) forcing fields are needed in order to remove UrbClim biases, so an RCM may still be needed to intermediate  between the GCM and UrbClim in cities like Barcelona. This needs to be explained in the manuscript, and we acknowledge that the overall discussion of the results needs to be improved.

*Finally, the authors do not provide any explanation of what processes are better represented in UrbClim that make it perform better than the regional climate model?*

Addressing biases in climate models, especially in complex models such as WRF, is difficult and seldom evaluation papers in the literature manage to offer a clear explanation of the biases found. Here we focus on the practical problem of producing high resolution simulations of the temperature at a city level, including the UHI. We show that a simplified approach, essentially running the UCM plus the PBL, produces comparable results to an RCM, using much less computational power, and reaching x4 resolution (250m Vs 1 km). We believe that is a result worth publishing, without needing to analyse in deep the biases found.

*In my opinion, the starting point is not correctly posed and the authors do not adequately support their conclusions with a rigorous analysis. I agree with the authors that for this particular application, UrbClim might present advantages over a Regional Climate Model coupled with an urban model, but I don't think the authors provided enough evidence for that.*

We think that the several parts of the manuscript need to be re-written in order to clarify the points that we commented above. Stating that UrbClim is "better" that the RCM in terms of performance is not the goal of the paper, and perhaps is going too far, given that the reanalysis run does have

biases comparable to WRF, or clearly larger in the case of the wind.

*In my opinion, the authors could make additional experiments, perform a like-to-like and more in-depth comparison, with possible reasons as to why UrbClim outperforms the RCM.*

We believe that additional experiments are not necessary, but a re-writing of some parts of the paper and clarifications of the points explained above. Also, as mentioned above, explaining WRF biases would require a separate study, and it is out of the scope of the present work.

*In that case, they should also mention that RCMs are a tool design to conduct atmospheric research and therefore have a wider range of applications, and this is the reason why they are selected over offline and faster models.*

This will be mentioned in the new version of the manuscript.

*As it is, I am unsure the paper makes a scientific or model development contribution worth publishing. Perhaps including the additional analyses suggested above could lead to a paper that is adding to the current knowledge. In addition to some general comments, I have also suggested some specific and technical comments aimed at improving a future version of the manuscript. Therefore, I would not recommend this paper for its publication at Geoscientific Model Development.*

We defend that the study presented in the paper is original and relevant in several ways: Geographical location, interesting climate (with a strong sea-breeze), length of the simulation (five months), that enables a robust evaluation, and comparison of UrbClim (with two driving models) and WRF. Thus, it is a contribution worth publishing. The reviewer mentioned some issues with our interpretation of the results that need to be addressed and clarified in the manuscript.

Answer to general comments:

1. We address this issue, these explanations mentioning the internal variability will be improved in the new version of the manuscript.

2. In the new version of the manuscript we will add more details about WRF configuration and to elaborate more on the differences with UrbClim. However, as mentioned before, model biases are usually hard to explain, so we are unlikely to be able to explain all the results in physical terms, as it is the case of the majority of climate model evaluation papers.

3. This is already considered, and will be highlighted more in the next version of the manuscript. However, note that the high computational cost of very high resolution RCMs is also a severe limitation for many applications.

4. The new experiments proposed by the reviewer would be very interesting. However, we believe that current results are interesting and clear enough for the paper to hold, provided that their explanation and interpretation is improved following reviewer's guidelines. The abstract will be modified too to change the sentence mentioned in the comment. As explained above, WRF is a mesoscale model, and with the domain size used it generates its own daily breeze cycle, as shown in figure 3 of the manuscript, so the comparison with UC-FC is not meaningless, though is very influenced by the small bias of ECMWF forecast model itself.

(there is no point 5)

6. Apart from the spatial coverage, UrbClim allows long multi-decadal adaptation experiments where changes in the city surface parameters can be tested (e.g., the color of the roofs). This would not be possible with a statistical model. The idea is to simulate only the fundamental processes that cause the Urban Heat Island, so the model is lightweight, but still based on physics, and thus allowing sensitivity experiments. Statistical downscaling is also very observation dependent, and has generalization problems when applied to future periods with climatologies far from the calibration period.

Specific comments will also be addressed in the new version of the manuscript, and answered in the final answer that will be submitted once the discussion is closed.

Best regards

The authors

---

## Short Comment (SC1) · 1 Apr 2016

General Comments:

The paper addresses advantages of using a fast Urban Climate Model (UrbClim) over a full scale mesoscale model (WRF) by using a case study depicting UHI simulation for the city of Barcelona. Urban Climate Model is relatively of high resolution that is driven by ERA interim reanalysis data or GFM of ECMWF. The model has earlier been validated for other European cities and now being implemented for Barcelona and utilized for climate projections. The model is demonstrated to be computationally efficient with higher resolution than a mesoscale model. However, as mentioned in the title, the model (UrbClim) is not a urban canopy model as canopy features are

not included in the true sense and the mesoscale model WRF does not include WRF-UCM. Hence the title needs to be appropriately modified with direct usage of UrbClim Though UHI is derived from the temperature differences, the paper presntlly lacks the robustness of estimating UHI. Further, model validation shall include meteorological parameters such as Wind speed and PBL etc. to demonstrate the efficacy of the two models. It would be interesting to see the performance of all the key meteorological parameters from the two models to examine the comparable performance and infer of the desired efficiency of UrbClim and this model comparison alone is adequate to make this point. The comments are elaborated below:

1. Line 90 onwards: Station number 6 is located on a rooftop at 33m while Station number 5 which is a rural station in located in a delta. The thermal properties of land surface and a land use predominently reflecting a water body is drastically different and the fact that no inversion is observed over rural station (considered here as reference station for UHI) would lead to erroneous representation of estimated UHI of the city. 2. Figure 1 a and 1 b: Figure 1 b should also depict the rural and urban stations. Both figures should depict coordinates (lat/long) using ArcGIS or another appropriate software. It appears that Urban and rural stations are not within the same nest and with similar resolution. This might affect the results somewhat. This needs to be clarified. 3. Brief description of incorporation of urban canopy model (physics/key eq.) in Urb-Clim model shall be included to explain the science and for comparison with mesoscale model to understand level of simplification. What are the time step for running these two models? A climate model run needs larger time step and a weather model like WRF requires lower time step to run. If this difference is large between the two models, UrbClim will obviously be computationally much more efficient. It would be more appropriate to compare UrbClim with another Climate model with similar or comparable time step resolution for depicting it's efficiency for predicting average temperatures from May to Sept. 2011. Essentially, this efficiency is one of the major claim by the authors in this paper and that is not shown by considering two dissimilar type of models in terms of phenomenal applicability of UHI. Further by including other meteorological

parameters such as wind speed, fluxes and PBL and demonstrating the comparative model performance would prove the point more effectively.

4. UHI requires detailed description of measurement sites including station pictures with surroundings. 2 station data will not be sufficient for robustness. Thus, it is suggested that model implementation and efficacy claim is limited to temperature predictions and other met. parameters shall also be included for this purpose. Further, authors claim that there are 11 met. stations in the domain; however data for only 2 stations is being used to depict UHI. Were the chosen stations showed the maximum UHI? How about UHI for other stations ? Could they reflect justifiable trends vis-a- vis their LULC? Most of the study would use 25 or more stations for UHI(Mohan et al., 2013:, Assessment of urban heat island effect for different land use–land cover from micrometeorological measurements and remote sensing data for megacity Delhi, Theoretical and Applied Climatology, 112, 647-658. DOI 10.1007/s00704-012- 0758-z ). On line 190, authors mention that " The measurement of the UHI with only two points has some limitations, as it may be sensitive to very local features such as the land use in the vicinity of the stations. However, the representativeness of these points has been carefully checked with high resolution satellite images". No details are provided as to how it has been checked. In addition, the agreement with previous studies increases our confidence in the results here presented. No details of any previous studies provided here. Satellite data would represent LST and not air temperature while UHI estimated and shown are based on air temperature.

5. Does the same trend in UHI was seen based on LST also? It is not clear. UHI phenomena occurs on a diurnal scale for which WRF-UCM is applied (Bhati and Mohan, 2015; TAC; doi:10.1007/s00704-015-1589-5). RMSE of temperature of about 2 deg. is acceptable as per WMO guidelines on shorter time scales of an hour or day as demonstrated in this study. Therefore results can be sufficiently robust for temperatures averaged over 5 months but the differences of 2.5 deg. as in UHI may not be ; hence model application seems appropriate for temperatures and not for UHI. More-

Interactive
comment

over, different heights of urban and rural stations will further add to this uncertainty. 6. Simulations are carried out from May to Sept, 2011. It is not clear for statistical paired analysis what temporal and spatial resolution and how many data points are used? 7. It will be good to include the monthly variations of UHI considering seasonality in May to Sept. data and examine the trends ? Similarly for the air temperatures and LST as well the monthly variations could be included. 8. Based on model simulations the spatial variation of UHI needs to be studied. It shall be shown whether spatial variation shows maximum difference at the two selected points and other places in the domain are depicting lower UHI so that urban -rural contrast can be deciphered. 9. The title mentions fast Urban Canopy Model while the abstract and text categorises this as Urban Climate Model and the mesoscale model used is WRF and not WRF-UCM. As per the text, WRF includes USGS LULC and no mention of WRF UCM is made. WRF has tremendous scope of improvement by using recent LULC other than USGS and including Urban Canopy as demonstrated by Bhati and Mohan (2015). Thus urban canopy model in the title may be replaced with urban climate model.

―――――――――――――――――――――

---

## Referee Comment (RC3) · Anonymous Referee #3 · 19 Apr 2016

The authors compare three different types of model runs for Barcelona. The details of the models are not provided. A summary Table which compares the key features (model characteristics, run resolutions, etc) and could include the computational resources difference, and key performance differences would be a useful addition. This could be cited throughout the paper (methods, results) to allow the reader to be clear how the benefits/costs are arrived at.

More details are needed on the measurements and processing of the evaluation data; the implications of the study period selected (clear). The comment (L200) concerning the gridpoints and the land use for the evaluation data needs to be made clearer or justified. It appears a better result is being selected – rather than understanding if

there is a larger issue.

All figure captions should be standalone. Add additional material/text to these.

Editorial comments – only one example given – correct throughout.

1. L5 use the term evaluated not validated (and equivalent throughout)

2. L8 including not using

3. L18 use the 'most well-known' rather than 'main'

4. L36 – reword

5. L41 – see point (1) (repeated through text)

6. L47 250 m, not 450m (change throughout)

7. L63 use 13.7 not 13,7 notation (correct throughout)

8. L65 reword

9. L70 – Figure not figure

10. L85,86 4, 7 – numbers less than 10 write in full

11. L88/9 – what height and exposure? How high is the sensor? Be clear about samples and averages.

12. L92 – Cereal fields, so changing height through the course of the year

13. L93 – how high above the roof? What is the height of the building?

14. L94 Km should be km

15. All maps need scales.

16. L105 on – what correction used for emissivity? Between areas/urban etc

17. L110 be clear that selection of no cloudy days introduces a bias to certain meteorological conditions

18. L160 cite chapter authors, not the book

19. Table 1 link to Figure 1 (stations); Define Variance ratio or cite reference

20. L118 meters -> UK or USA English?

21. L170 standard scores or metrics – reference

22. L170 2 m

23. Figure 2 – indicate in the caption where codes for key are explained. Captions should be standalone

24. L185 Check in all places oC; or express in terms of K and remove o. In some places reversed °C (e.g. Table 1)

25. L189 – Garcia

26. L193 - Note importance of land cover. What do they represent in terms of Local Climate Zones?

27. L193 cite these previous studies

28. L200 on – But don't you need to check all grids now? Land use? Advection? Etc. what

29. Figure 3 – be explicit about UHI – temperature difference

30. L 222 (e.g. Figure 4) rather than which are here depicted in . . ..

31. L231 Figure 5 (introduce space)

32. Figure 4 – Significant figures! Relabel X-axis no need for May 2011 on all as in caption

33. L234 as above space between number and units

**[GMDD](GMDD)**
34. L240 as above evaluation rather than validation

35. L 242 – typos near delete .advection

36. L256 – be more explicit about long spin-up. Some suggest for certain models 10-20 years are needed to get soil moisture characteristics correct.

37. Line 319 data were not was

---

## Author Response (AR1)

**Answer to reviewers**

Author comments are in black and reviewer comments in blue.

**Anonymous Referee #1**

Received and published: 3 March 2016

"general comments"

This paper adresses interesting results on the scope of the presented topic. The methodology is clearly described and the results support the reached conclusions. From my point of view, the manuscript has the quality required to be published after some minor technical corrections. However, I missed some more details on the analysis of results showed in figure 6 (see "specific comments").

Authors would like to thank RC1 for this positive review.

**"specific comments"**

Line 97-98: I disagree with this argument: "the surrounding topography is flat, which does not favour temperature inversions during night time hours". In fact, flat terrain favours the development of a nocturnal stable layer. However, since inversions are more related on atmospheric stability, I suggest to change this argument. Perhaps you just can comment that there are no significant orographic objects between the two stations.

The sentence has been re-written following the reviewer's suggestion. The new sentence reads as follows: The rural station is located in a delta, and therefore the surrounding topography is flat, with no relevant orographic objects between the two stations.

Line 239-240: Do you think that the LST overestimation of both UrbClim and WRF can be related to the fact that the cloudy nights are not considered after filtering the satellite images as it is described in chapter 2.2? In other words, to compare both models against satellite, the same days have been considered to compute the LST averaged during nightime hours for the warm season (figure 6)?

The days (and the times of the day) considered in the averages are the same in the model data and in the observations. The 8-day averages flagged as containing cloudy days in MODIS were masked before computing the averages in both sides. We devoted a lot of effort to review the code that carries out this calculation, which is not trivial, in order to be sure that the comparison was performed under the same conditions and after checking for errors in the codes. Thus, the cause of the LST overestimation must be a different one. According to some calculations we did, different emissivity used by MODIS and the models can explain up to 50% of the bias. The remaining 50% may be related to soil parameters or small differences in the definition of the surface in the urban environment. As the LST is a very sensitive variable, there are many possible causes that could potentially explain the bias, and therefore we prefer to investigate in depth this interesting issue in another article.

Line 287: I think it is important to emphasize that the first conclusion is obtained after analysing observational data and valid only for the warm season.

This is emphasized in the new version of the manuscript: *"The average UHI in the city of Barcelona during the warm season (May-September) reaches 2.5°C at night."*

Line 93: style. Substitute "...at the city centre of the city..." for "...at the city center of Barcelona..."

The sentence has been modified as suggested.

Line 146: typing error. Substitute "...important give the..." for "...important given the..."

The typo has been corrected.

Line 154: typing error. Substitute "...nested domains 1b..." for "...nested domains (figure 1b)..."

The typo has been corrected.

Line 242: typing error. Remove "advectionAs"

The word "advection" has been removed, thanks.

**Anonymous Referee #2**

The authors would like to thank again the referee for the thorough review. The main comments were already addressed in the interactive discussion. Some of these answers are here reproduced for clarity, together with references to the changes implemented in the manuscript to address the reviewer's concerns.

This paper is aimed at showing the advantages of an off-line urban canopy model with respect to a regional climate model coupled to an urban canopy model for urban heat island studies.

We would like to note that UrbClim is not only an offline urban canopy model (UCM), but it parametrizes the planetary boundary layer (PBL) and performs a simplified simulation of the lower part of the atmosphere (see section 2.3 of the manuscript, and De Ridder et al. 2015). In order to stress these points, the following changes were introduced in the manuscript:

- The title has been changed, and now it reads: "Advantages of using a fast urban boundary layer model as compared to a full mesoscale model to simulate the urban heat island of Barcelona"

- In the abstract, UrbClim is more specifically defined as an "urban boundary layer" model, and not just as an urban climate model.

- In the introduction, it is now explained that UrbClim simulates both the PBL and the surface: "Here we show how, by using a simplified model that only accounts for the Planetary Boundary Layer (PBL) and the surface physics, it is possible to reach resolutions of 250 m with affordable computational resources".

It is undeniable that there is benefit in the use of application-specific models such as UrbClim, particularly in terms of computational costs. In such circumstances, even if they both show similar results, the computational costs associated with the regional climate model make the use of UrbClim very attractive.

The computational cost is not just an added value, but a critical factor, because it allows the performance of multi-decadal simulations in large ensembles of cities at sub-kilometer resolutions. The cost of doing this with a Regional Climate model (RCM) is extremely difficult to afford. Note that the computational cost is in turn the very reason of existence of RCMs.

However, in order to justify the use of a standalone urban model, the authors compare completely different tools (i.e. a more like-to-like comparison would be to show benefits of UrbClim over the Single-Layer Urban Canopy Model, both running offline with the same boundary conditions) and do not fully acknowledge (only at the very end of the paper) that a regional climate model coupled with an urban model incorporate features that an offline model cannot (e.g. two-way interaction with surrounding circulation such as sea-breeze).

As we pointed out, UrbClim is not exactly an offline UCM. The paper aims to compare UrbClim with a widely used RCM such as WRF, and not to compare UrbClim with the UCM of WRF. The scope of the paper is applied and pragmatic, in the sense that it is not trying to find the best theoretical approach to model the Urban Heat Island (UHI). We try to show that in this case, it is not necessary to run the primitive equations in the whole troposphere, together with the parameterizations of radiation, cumulus... etc., as WRF does, in order to reproduce a realistic UHI. Instead, we show that UrbClim, which is a simplified model that only accounts for the Planetary Boundary Layer (PBL) and the surface physics, simulates the UHI of Barcelona with comparable skill to a full mesoscale model. We think that this is a strong practical result, which justifies the use of UrbClim in for the study of specific aspects of the urban climate.

**But what is more important, the comparison is to a large extent unfair because the authors claim a better representation of local temperatures when the Urb-Clim is driven by ERA-Forecast, but do not test the regional climate model running with the same large/mesoscale information.**

This seems to be a major concern for the reviewer. A fair comparison between two completely different approaches is not fully possible. A regional model like WRF is able to generate his own mesoscale variability so, in principle, it should be able to compete with the UrbClim run nested in ECMWF forecast model, as the large-scale variability is very similar in the two global models used.

In fact, it could be argued that WRF has an advantage here, as its UCM and PBL are running with forcing fields at a 15 times better resolution than those used for UC-FC. These kind of issues make the fair comparison between the two approaches complicated (though the comparison is certainly not fair for the WRF UCM alone, but as mentioned, this is not the goal of the paper). The following paragraph has been added to the manuscript to explain this point (L 204-276):

"From these results, it is clear than the higher resolution of the ECMWF forecast model respect to ERA-Interim (16 km vs 70 km) is greatly improving the performance of UrbClim. It is however unclear how to understand the comparison of WRF with UrbClim. On one hand, UC-ERA and WRF, which are both nested in the same reanalyses, display overall similar scores in temperatures, and WRF does better in the wind speed. However, WRF carries on a full dynamical downscaling up to 1.1 km resolution so, in principle, it should be able to achieve an accuracy similar to UC-FC. But the UrbClim run nested in the ECMWF forecast model does show slightly better scores than WRF. Given the large number of factors involved, it is difficult to find an explanation to this result in physical terms. In general, WRF is more biased than UC-FC (figures 4, 5 and 6) and than the ECMWF forecast itself (not shown). It produces too cold temperatures and slightly too high wind speed during the day. As WRF is very customizable, it may be possible to remove these biases with a more careful configuration. However, WRF biases in the wind speed have been proven difficult to correct, and research is yet ongoing in this line (García-Díez et al., 2015; Lorente-Plazas et al., 2016)." The problem here may be that WRF is more biased than the ECMWF forecast model, which is stunningly precise. So, that is why we say in the conclusions (L310-312): "*From these results, it is reasonable to infer that the skill of UrbClim, and probably of other similar urban climate models, is constrained by the performance of the driving model, and particularly for variables that are important for the UHI, this is, wind speed and cloudiness*". This holds also for WRF-UCM. If driven by ECMWF forecast model, WRF-UCM may be able to perform as well as UC-FC. But testing this is out of the scope of the paper.

There may be other RCMs showing better results but, as WRF is currently widely used in this kind of studies about the UHI effect, we think that the results of the paper are relevant. On the other hand, results show that relatively high resolution (16 km) forcing fields are needed in order to remove UrbClim biases, so an RCM may still be needed to intermediate between the GCM and UrbClim in cities like Barcelona. This needs to be explained in the manuscript, and we acknowledge that the overall discussion of the results needs to be improved.

**Finally, the authors do not provide any explanation of what processes are better represented in *UrbClim that make it perform better than the regional climate model?**

Addressing biases in climate models, especially in complex models such as WRF, is difficult and seldom evaluation papers in the literature manage to offer a clear explanation of the biases found. Here we focus on the practical problem of producing high resolution simulations of temperature at a city level, including the UHI. We show that a simplified approach, essentially running the UCM plus the PBL, produces comparable results to an RCM, using much less computational power, and reaching x4 resolution (250m vs. 1 km). We believe that is a result worth publishing, without having to analyse in depth the biases found.

In my opinion, the starting point is not correctly posed and the authors do not adequately support their conclusions with a rigorous analysis. I agree with the authors that for this particular application, UrbClim might present advantages over a Regional Climate Model coupled with an urban model, but I don't think the authors provided enough evidence for that.

We have rewritten several parts of the manuscript in order to clarify the points commented above. The article does not claim anymore that UrbClim is "better" than WRF in terms of performance, given that it is not the aim of the paper. But at least we think that results here shown prove that the performance of UrbClim has similar skill than that of WRF, under comparable conditions. To prove that, we show that UrbClim and WRF, both driven with Era-Interim, have similar skill in the simulation of the temporal and spatial patterns of the urban climate in Barcelona. As an additional result, we also show that the performance of UrbClim can largely improve when the Integrated Forecast System is used as a driving simulation, although this is not part of the direct comparison with WRF.

**In my opinion, the authors could make additional experiments, perform a like-to-like and more in-depth comparison, with possible reasons as to why UrbClim outperforms the RCM.**

We thank the referee for this suggestion, we however think that additional experiments are not necessary, given that the direct comparison between UrbClim and WFC is shown by means of the use of ERA-Interim as the driving simulation. Instead, we have rewritten parts of the paper and clarified the points explained above. In this way, we do not claim that UrbClim outperforms the RCM, but instead we show that they have comparable skill. As an additional result, we also show that the performance of UrbClim can largely improve when the Integrated Forecast System is used

as a driving simulation, although this is not part of the direct comparison with WRF.

In that case, they should also mention that RCMs are a tool design to conduct atmospheric research and therefore have a wider range of applications, and this is the reason why they are selected over offline and faster models.

The objective of the paper is to show that, despite the simplifications of UrbClim, its simulation is not worse than the RCM, and therefore, given its lower computational requirements, it can be used as an alternative tool to model some urban climate processes, in this case, in the city of Barcelona. Nonetheless, it is not our aim to perform a detailed study of the dynamics of the whole troposphere, given that UrbClim is a simplified model of the lower troposphere.

As it is, I am unsure the paper makes a scientific or model development contribution worth publishing. Perhaps including the additional analyses suggested above could lead to a paper that is adding to the current knowledge. In addition to some general comments, I have also suggested some specific and technical comments aimed at improving a future version of the manuscript. Therefore, I would not recommend this paper for its publication at Geoscientific Model Development.

We think that the paper is original and relevant due to the geographical location of the study, the complex climate here studied (with a strong sea-breeze system), and the length of the simulation (five months). We think that it is also an important contribution worth publishing because it enables a robust evaluation and comparison of UrbClim (with two driving models) and WRF.

The reviewer mentioned some issues with our interpretation of the results that have been addressed and clarified in the new version of manuscript.

Answer to general comments:

1. The authors mention internal variability in multiple occasions, but it is unclear from their discussion what they mean by internal variability. It is also unclear why internal variability is regarded as intrinsically negative (L161-165). I understand that the authors preference is to avoid model departures from the boundary conditions in terms of the large scale (or mesoscale) conditions and I agree that in that sense, UrbClim does not generate any "internal variability" (L178), but in a broader sense, it is required that models produce some internal variability so they produce results the are different from the boundary conditions (added information). In any case, the authors need to be more explicit about what they mean by "internal variability". The authors also mention internal variability to justify the resolution jump between the boundary conditions and the UrbClim resolution (L121-123). I don't quite understand the sentence and why it is acceptable to have such a difference between resolutions. UrbClim is effectively downscaling a single grid point of ERA-Interim and therefore forced everywhere with the same conditions. This is something that compromises the representativeness of the results.

We agree with the reviewer, and therefore we have improved the discussion about internal variability throughout the paper. We believe this clarification is key for the improvement of the interpretation of the results. The modifications are:

**Section 2.3, (The UrbClim Model)**

"Mesoscale models are tied to their driving models by the boundary conditions. Yet, they develop internal variability (Giorgi and Xunquiang, 2000). In the case of UrbClim, the small size of the domain, and the simplicity of the atmospheric component, greatly reduce the internal variability. The model can be seen as a "wind tunnel". This can be a limitation for the capability of the model to add value to the coarser resolution boundary data, especially for variables like wind.

On the other hand, the small internal variability has also advantages, as the stability of the simulation is not compromised by the difference in the resolutions of the UrbClim and driving models, as it normally occurs with conventional mesoscale models. Nonetheless, this resolution jump can sometimes affect the quality of the simulation if the driving model does not accurately reproduce the local climate. This balance between the internal variability and the computational power will be key for the interpretation of the results."

2. What does UrbClim do better than WRF-SLUCM? It is necessary that the authors provide a better description of the models and a reasoning of which processes UrbClim might be improving. It is necessary to describe how are the city characteristics seen by the models, what are the differences in the two models (e.g. extension, density, types of building, vegetation cover. . .) Are they different in each of the models? What are the differences how UrbClim deals with urban and rural areas? And WRF-SLUCM?

In the new version of the manuscript we added more details about the configuration of WRF and UrbClim, which help to understand the basic differences between both models. Also note that we included two new figures (figures 2 and 3) with the land cover used by UrbClim and WRF, mapped from CORINE in both cases.

UrbClim and WRF-SLUCM are described in detail in the references provided. Additionally, a new paragraph has been introduced at the end of section 2.3, briefly describing the differences between the urban canopy models of WRF and UrbClim.

Regarding the comparison between UrbClim and WRF-SLUCM, as mentioned before, model biases are usually hard to explain, so we are unlikely to be able to explain all the results in physical terms, as it is the case of the majority of climate model evaluation papers. However, in the new manuscript, we do not claim anymore that UrbClim is better than WRF, but instead we say that they show comparable skill. The abstract now says: *"The results show that, generally, the performance of the simple model on reproducing the Urban Heat Island is comparable to the mesoscale model."*

3. The extension of UrbClim domain is roughly 25 km by 25 km. UrbClim is essentially an offline model driven by 1-grid point from ERA-Interim and 4 grid points in ERA-FC. It is obviously much faster than the RCM (considering the number of grid points, including the vertical, in each model provides an idea of the considerably lower computational requirements of UrbClim. This is a very important feature, but it should be considered that the area simulated is much smaller and the applications more limited.

This is already considered, and it is highlighted in the next version of the manuscript. However, we would like to note that the high computational cost of very high resolution RCMs is an important limitation for many applications related to the urban climate. Namely, those needing multi-decadal simulations with large ensembles of models like CMIP5 models in order to account for uncertainty. Note also that, while the applications of UrbClim are certainly more limited than RCMs, a large number of urban climate studies focus precisely on the heat stress and the UHI, so they fall into the scope of UrbClim.

4. The comparison is unfair in many different ways. UC-FC and WRF cannot be compared in any way (They are forced by different mesoscale conditions). In the abstract, the author anticipate a better performance of UC-FC, but in my opinion this result does not prove that UC outperforms the RCM. The authors are evaluating at the same time the performance of ERA-Interim and ERA-FC,

and UrbClim and WRF. This comparison would only be acceptable if WRF is driven by ERA-FC too. I would suggest the authors consider other experiments to adequately assess the performance of the models. In addition to WRF driven by ERA-FC, UrbClim could be driven by WRF outputs and test if it improves WRF estimates.

The new experiments proposed by the reviewer would be very interesting. However, we believe that current results are interesting and clear enough for the paper to hold, as their explanation and interpretation has been improved following reviewer's guidelines. The abstract has been modified too to change the sentence mentioned in the comment. As explained above, WRF is a mesoscale model, and with the domain size used it generates its own daily breeze cycle, as shown in figure 3 of the manuscript, so the comparison with UC-FC is not meaningless, though is very influenced by the small bias of ECMWF forecast model itself.

Nevertheless, in the new version of the manuscript is is clarified that UC-ERA is the simulation truly comparable with WRF. UC-ERA is now included in the figure showing the minimum temperature climatology, and is the one used in the LST comparison (figures 7 and 8 in the new manuscript), instead of UC-FC. The skill and bias of UC-ERA is clearly comparable to WRF, so the message of the new version of the paper is well supported.

**(there is no point 5)**

6. The question that arises after reading the manuscript is, why not using statistical downscaling, which is even faster, allows for large ensembles and will reproduce present climate results much better as the are purposely calibrated. Why would an user opt for UrbClim over statistical downscaling? I would guess the answer is the spatial coverage of UrbClim.

Apart from the spatial coverage, UrbClim allows long multi-decadal adaptation experiments where changes in the city surface parameters can be tested (e.g., the color of the roofs). This would not be possible with a statistical model. The idea is to simulate only the fundamental processes that cause the Urban Heat Island, so the model is lightweight, but still based on physics, and thus allowing sensitivity experiments. Statistical downscaling is also very observation dependent, and has generalization problems when applied to future periods with climatologies far from the calibration period. In the particular case of Barcelona, the scarcity of station data would be a strong limiting factor for statistical downscaling.

**Specific comments**

L44 Regional climate models have multiple applications, not just downscaling climate projections. Perhaps: "RCMs are limited area models used to downscale climate change projections from coarse resolution Global Circulation Models as well as other applications."

The sentence has been modified as suggested.

L53 Please specify which of the Urban Canopy options if the authors want to keep the following sentence. "This parameterisation. . .". Otherwise, specify only in the methods. Also, if the authors refer to Single-Layer Urban Canopy Model, it was developed by Kusaka et al. (2001). Chen et al. (2011) describe the implementation in WRF. I also miss some references to work done with WRF/SLUCM for future projections (e.g. Georgescu et al. 2013, Argueso et al. 2015, Kusaka et al. 2012).

A new sentence has been included to clarify that the parameterization used is the Single-Layer

Urban Canopy Model. Kusaka et al. (2001) is now cited, and we also mention three papers about future projections.

L64 Add ", especially" after urban pollution. Urban pollution exacerbates sensitivity to adverse conditions is for all population segments.

The sentence has been modified as suggested.

L70 (FIGURE 1) I would be necessary to have some sort of reference of the urban extension in Figure 1. For those of us not familiar with the region it is difficult to locate the city, and urban areas in general, in the map. It is a major feature of this study and should be shown.

Instead of doing this, two new figures with the land use classes used by UrbClim and WRF where included in the paper (figures 2 and 3). This is related to requests by reviewer#3 and we believe that it helps to understand the interpretation of the results.

L93 How representative is the temperature in the roof of a building of the temperature at levels that matter for population? How comparable is that to the temperatures provided by both models at 2 m .

This is an accurate assumption. See this paragraph of De Ridder et al (2015) and the references:

"This homogeneous mixing assumption in the urban canopy layer is supported by several studies. Nakamura and Oke (1988) measured very slight air temperature gradients only through most of the urban canopy layer, as long as the considered location was not too close to a lateral surface. In their urban canopy model, Erell and Williamson (2006), make this same assumption. Also, measurements acquired in a street canyon in Basel (Switzerland) at different times of the day (Rotach et al., 2005) show little vertical temperature variation, and show that the top-of-canopy temperature is fairly representative of values lower down in the canopy. Finally, in a numerical experiment conducted with a computational fluid dynamics model on an idealized street canyon, Solazzo and Britter (2007) found the canyon air temperature to be spatially nearly uniform, apart from a thin near-wall thermal boundary layer."

Erell, E., Williamson, T., 2006. Simulating air temperature in an urban street canyon in all weather conditions using measured data at a reference meteorological station. Int. J. Climatol. 26, 1671–1694.

Nakamura, Y., Oke, T.R., 1988. Wind, temperature and stability conditions in an E–W oriented urban canyon. Atmos. Environ. 22,2691–2700.

Rotach, M.W., Vogt, R., Bernhofer, C., Batchvarova, E., Christen, A., Clappier, A., Feddersen, B., Gryning, S.-E., Martucci, G., Mayer, H., Mitev, V., Oke, T.R., Parlow, E., Richner, H., Roth, M., Roulet, Y.-A., Ruffieux, D., Salmond, J.A., Schatzmann, M., Voogt, J.A., 2005. BUBBLE – an urban boundary layer meteorology project. Theor. Appl. Climatol. 81, 231–261.

Solazzo, E., Britter, R.E., 2007. Transfer processes in a simulated urban street canyon. Bound.-Layer Meteorol. 124, 43–60.

L97-99 I disagree with this statement. Flat areas are indeed characteristics of temperature inversions, particularly near the coast. Please revise. The sentence has been re-written following the referee's #1 suggestion. The new sentence reads as follows: "The rural station is located in a delta, and therefore the surrounding topography is flat, with no relevant orographic objects between the two stations".

L112-113 In the estimate of missing values, did the authors considered all land points within the UrbClim domain? Why 14%?

All land points within the UrbClim domain were considered. The 14% threshold was chosen after inspection of the data, seeking for a compromise between the threshold and the number of days of data finally considered. Lower thresholds lead to the rejection of most of the days.

L116 What to the authors mean by "main features of the urban climate"? Are there hydrological variables (e.g. precipitation, evaporation).

In the model output there is evaporation, together with the latent and sensible heat fluxes, but no precipitation (it is read from the driving model). The sentence has been modified to make it more specific: *"The UrbClim model is designed to simulate the temperature and heat-stress fields at a city scale requiring the minimum amount of computational power, so that it is possible to perform long runs at a resolution of hundreds of metres."*

L132. In which ways? If the extension is not described, this sentence can be removed.

The description has been expanded as requested by referee #3.

L164-165 It is not clearly described how this configuration deals with soil variables. Are they obtained from ERA-Interim at every reinitialisation? If so, NOAH LSM is constantly trying to balance the information from ERA-Interim LSM and they are not necessarily compatible (the even don't share the same layers). A similar question arises for Urb-Clim in terms of how it sees soil temperature and moisture.

In the case of UrbClim, the simulation is initiated from ERA-Interim and run continuously. In the case of WRF, all the fields, including soil moisture and temperature, are restarted daily, leaving 12h of spin-up. The soil variables are interpolated from the ERA-Interim soil levels to the WRF and UrbClim soil levels by the respective preprocessors.

As the reviewer says, the re-forecast running scheme used to run WRF can in principle introduce a bias if the soil parameterizations are very different. However, the long time it takes to do a full soil spin-up (about one year) means that the soil state related model drift is negligible in short 36 hours simulations, so the physical consistency is granted. In the case of this paper, WRF cold bias during daytime hours seems more related to the overestimation of the wind speed and thus the sea breeze, than to a soil-moisture and evaporation overestimation. Also, note that some studies have found that the soil spin-up can sometimes reinforce the bias rather than removing it (see the comparison between REFOR and MPE-G in García-Díez et al., (2015)). Yet, we agree with the reviewer in that this topic is something worth to be studied with more detail in future experiments.

García-Díez, Markel, Jesús Fernández, and Robert Vautard. 2015. "An RCM Multi-Physics Ensemble over Europe: Multi-Variable Evaluation to Avoid Error Compensation." *Climate Dynamics*, February, 1–16. doi:10.1007/s00382-015-2529-x.

**Table 1 How is the variance ratio calculated? What does it represent?**

Variance ratio is the quotient between the variances of the model and the observation, i.e var(model)/var(observation). Values above 1 indicate that the model produces too much variability,

and values below 1 the opposite.

L170-174 Do these values in UC-ERA ultimately depend on information provided by a single grid point from ERA-Interim? The resolution of ERA-Interim is equivalent to  $\sim$ 70 km, while the extension of the UrbClim domain is  $\sim$ 25 km by 25 km.

Essentially yes, that is why we use the analogy of UrbClim being a sort of "wind tunnel", as the boundary conditions are the same in both sides of the domain, but it does use also a pressure gradient term, computed with the surrounding gridpoints.

L179-180. It is not correct to say that the "results can be interpreted as a comparison between Urban Canopy+PBL models driven by" different boundary conditions. In WRF there is a two-way interaction, the results from Urban Canopy+PBL are fed back into the dynamical core and influences local circulation (and potentially larger scale circulation).

Please see below.

L181-182 is especially concerning since the authors disregard that this comparison is mixing multiple things, the mesoscale information to begin with. I agree that at these scales (15km to 1km) and for variables such as temperature, there is little benefit in increasing resolution, but this cannot be inferred from the authors results.

This paragraph (L179-182) has been removed in the new version of the manuscript. It has been replaced by a new paragraph at the end of section 3.1. In this new paragraph, we discuss the difficulty of comparing WRF with UC-ERA and UC-FC, given the scope and resolution differences, in the line of the comments of the referee #2 and the answers given in the present document.

**L197-198 Is this a merit of FC over ERA-I rather than UrbClim?**

The improvement between UC-ERA and UC-FC is obviously a merit of FC, but the small bias of UC-FC is also a merit of UrbClim. Note that FC still does not resolve the city, and is more biased than UC-FC when comparing it to the station data (it does not see the UHI). See the figure below. Thus, we see that provided with the accurate wind of FC, UrbClim is doing its part of the job very well.

Figure 1: Observed (OBS) and modeled (ECMWF fc) 2m temperature daily cycles for the rural (left) and the urban (right) stations. The model is the ECMWF forecast model with aprox. 15 km resolution.

L200-205 It is obvious that if the model was assigning urban land use to the rural location, then for the purpose of this study they are not comparable by any means. I do not think this needs to be mention in the text. Instead, I would say that the closest grid point with a particular land cover (and specify the land cover) was selected to compare with the rural station, and perhaps say the distance between the grid point and the station.

The paragraph has been re-written, but yet the process is explained with some detail, to take into account also referee #3 comments. Also, a figure with the location of the station, WRF land use and the chosen gridpoints has been added as a supplementary figure, as requested by the reviewer #3.

L 206-207 I would say that the bias occurs throughout the day except in the evening (16H-00H)

The sentence has been changed as suggested.

L 210 This is not surprising. As the authors suggest, at 70 km the intensity of the sea breeze is often underestimated. But also, only information from 1 ERA-Interim grid point is provided to UrbClim.

We agree with the referee, although as mentioned above, also the pressure gradient is used, involving information from the surrounding gridpoints. However, the pressure gradient related to the sea-breeze in ERA-Interim is too small.

Figure 3. A physical explanation of why sea breeze is weaker in urban points would be desirable. It cannot simply be increased drag because the urbanised areas along the coast act as a barrier for the rural station too. Furthermore, the urban station is located in a roof, so depending on the surrounding buildings and the height of the station, the drag could be negligible for that station. In this case, the comparison tell us about the boundary conditions, not necessarily the UrbClim. It could be well the case that UrbClim is doing a fantastic job in both cases, but ERA-FC information is more accurate (or more comparable to those scales).

The shadow casted by the urbanized areas in the wind speed field is very short-lived, as the vertical mixing transports linear momentum from upper levels quickly after the flow leaves the urbanized area. Both WRF and UC-FC show weaker winds in the urban location consistently with the observations. Also, the Urban-Rural wind speed difference is smaller in UC-ERA because the wind is way weaker in general. Regarding the location of the urban station in a roof, this could be an issue if the building was higher than the surroundings, but it is not, and in thus representative of wind speed in the urban canopy.

L221-228 It is unclear what the authors want to illustrate with figure 4. What is the contribution to the paper? Isn't this day-to-day variability highly influence by the boundary conditions?

It was somewhat surprising to find that the UC-ERA run was getting the average UHI daily cycle right despite miss-representing the sea-breeze cycle. In the figure we show that, despite getting the average values right, this simulation is suffering from large errors in some days. This is fixed both in UC-FC by using higher resolution forcing, and in WRF by doing an actually complete dynamical downscaling. This figure also shows that the day-to-day variability of the UHI of Barcelona is significant, being able to reproduce it in the simulations is a way to improve heat stress forecasts in

the city.

L 230-238 (Description of figure 5) This is basically qualitative. It does offer finer detail because it has finer resolution, whether this is correct remains unclear, even after comparison with relatively low resolution MODIS (Figure 6), where a more quantitative measure is provided.

The goal here is to show that both UrbClim and WRF produce consistent spatial patterns despite being very different approaches, and also to show the extra detail that the 250 m resolution can offer. It is true that this detail cannot be properly evaluated because of the lack of observations, but it is plausible and based on physics.

L 239-240 (and onwards) Among the multiple descriptions of UHI, two are widely used. The skin temperature (or surface temperature) UHI and the screen level temperature UHI. Although linked, they involved completely different processes. Indeed, skin temperature UHI is generally positive at day and night, but the screen level temperature UHI is often negative during the daytime. The authors should capture this in their discussion.

A new sentence has been introduced at the end of the section to mention this: "*Finally, as mentioned in the introduction, LST and Surface Air Temperature (SAT) Urban Heat Islands are not equivalent, and are driven by different phenomena. Thus, it is also possible that models that reproduce the SAT UHI correctly produce a biased LST UHI.*"

L 241 I understand that results from Zhou et al. 2015 contrast with the authors finding rather than agreeing.

Yes, this is correct, we admit that the "also" used in the manuscript was confusing. In the new version it has been removed, and the sentence now reads: "*Other studies (Zhou et al., 2015)* found *small errors when comparing MODIS and UrbClim LST...". Thanks.*

L 249-250 If the authors provide these confidence bounds, both values are exactly the same (not just statistically insignificant). Not only the confidence bounds overlap but the both include the estimate from the other model.

This is mentioned in the text: "Thus, UrbClim correlation is higher, but the difference is not statistically significant, as the confidence bounds overlap."

**L 300 But UrbClim does not provide rainfall at all, or does it?**

That is correct. That is why WRF provides more detail. In the new version of the manuscript a more detailed description of UrbClim shortcomings is included.

**Technical corrections**

L34-36 Please revise sentence, how it links to the previous one. Please revise use of commas.

We followed the suggestion, removing a comma, a linking word and re-arranging the sentences:

"They found that the relative contribution of these factors depends on the local background climate of the city and on the time of the day. In general, during daytime, convection efficiency and evapotranspiration are the main drivers of the UHI, while heat storage is the most relevant during the night. Zhao et al. (2014) used satellite retrieved land surface temperatures, but these can defer from screen level temperatures. Other authors (Arnfield, 2003), highlight the complexity of the problem of measuring the UHI, because of the difficulty of getting observations of the urban climate with enough detail and reliability."

L37 Replace "defer" with "differ"

This mistake has been corrected as suggested.

L38-39 Please revise use of commas.

A comma has been removed: "*Furthermore, the complexity of the urban surface, featuring anisotropy and vertical surfaces, makes it complicated to sample by satellites (Voogt and Oke, 1998).*"

L41 The end of this sentence is unclear. Please rewrite. (The last half of this paragraph needs to be clarified)

It has been divided into two sentences to make it clearer: "These difficulties with the observations increase the value of numerical simulations, that can produce detailed fields which are not observable. At the same time, the lack of observations hampers the evaluation of these simulations."

**L59 Remove "now" (?)**

The sentence now reads: "Taking into account that the Mediterranean countries are currently more vulnerable to environmental summer conditions than other European societies, the larger magnitude of the projected temperature increase is expected to become a major challenge for public health (Ostro et al. 2012)".

Ostro B, Barrera-Gómez J, Ballester J, Basagaña X, Sunyer J. The impact of future summer temperature on public health in Barcelona and Catalonia, Spain. International Journal of Biometeorology 56, 1135-1144 (2012).

L93 Replace "Km" with "km"

This mistake has been corrected as suggested.

**L152 Remove comma after sub-modules. Missing verb before available?**

One particularity of this model is that i has a large amount of parameterization schemes, dynamical options and sub-modules, available to the user to choose among them.

**L242 Please remove "advection"**

This mistake has been corrected as suggested.

**Anonymous Referee #3**

The authors would like to thank referee #3 for the useful review.

The authors compare three different types of model runs for Barcelona. The details of the models are not provided. A summary Table which compares the key features (model characteristics, run resolutions, etc) and could include the computational resources difference, and key performance differences would be a useful addition. This could be cited throughout the paper (methods, results) to allow the reader to be clear how the benefits/costs are arrived at.

As requested by the reviewer, a table has been added to the paper (table 2) with relevant information about the model configuration (domain size, time step) and the results of the benchmarking experiment, and it is cited in the text. Model performance against observations has not been included in the table, because of the difficulty of finding single numbers representative of the model skill. This could lead to misunderstandings because, as referee #2 pointed out, the interpretation of the results in terms of "which model is the best" is not straightforward. Please note also that the models are described with great detail in the references.

More details are needed on the measurements and processing of the evaluation data; the implications of the study period selected (clear). The comment (L200) concerning the gridpoints and the land use for the evaluation data needs to be made clearer or justified. It appears a better result is being selected – rather than understanding if there is a larger issue.

In the new version of the manuscript, more detail has been introduced in the description of the evaluation data in section 2.1. This includes new supplementary figures that show clearly how the choice of the representative gridpoints in the WRF grid is justified.

**All figure captions should be standalone. Add additional material/text to these.**

The captions have been extended, following the reviewer's request.

**1. L5 use the term evaluated not validated (and equivalent throughout)**

All the forms of the verb "validate" have been replaced by "evaluate" throughout the paper.

**2. L8 including not using**

The sentence was left unchanged, as in WRF using an UCM or not is left at choice of the users.

**3. L18 use the 'most well-known' rather than 'main'**

The sentence has been modified following the correction.

**4. L36 – reword**

Referee #2 also commented on this part. The revised sentences are now: "They found that the relative contribution of these factors depends on the local background climate of the city and on the time of the day. In general, during daytime, convection efficiency and evapotranspiration are the main drivers of the UHI, while heat storage is the most relevant during the night. Zhao et al. (2014) used satellite retrieved land surface temperatures, but these can defer from screen level temperatures. Other authors (Arnfield, 2003), highlight the complexity of the problem of measuring the UHI, because of the difficulty of getting observations of the urban climate with enough detail and reliability."

**5. L41 – see point (1) (repeated through text)**

As mentioned, the correction of pint (1) has been applied in all the paper.

6. L47 250 m, not 450m (change throughout)

This correction has been throughout the paper too.

7. L63 use 13.7 not 13,7 notation (correct throughout)

The mistake has been corrected throughout the sentence.

**8. L65 reword**

The sentence now reads: This larger sensitivity to environmental conditions is exacerbated by urban pollution especially in old people living in cities with pre-existing or chronic cardiovascular and respiratory diseases (McMichel et al., 2006).

**9. L70 – Figure not figure**

This mistake has been corrected in new version of the manuscript.

**10. L85,86 4, 7 – numbers less than 10 write in full**

These numbers have been written in full as requested.

11. L88/9 – what height and exposure? How high is the sensor? Be clear about samples and averages.

The stations follow the WMO standards. Apart from that, we lack more precise metadata about sensor height.

**12. L92 – Cereal fields, so changing height through the course of the year**

The cereal fields surround the station, but the station location itself is not cereal, but grass and is regularly maintained, which includes cutting the grass.

**13. L93 – how high above the roof? What is the height of the building?**

See the new information added in the manuscript.

**14. L94 Km should be km**

This mistake has been corrected in new version of the manuscript.

**15. All maps need scales.**

The software used for producing the maps does not allow to plot scales when using the regular lonlat (Plate Carree) projection. Instead, the axes have been labeled with latitudes and longitudes, which add information both about the scale and location of the maps.

**16. L105 on – what correction used for emissivity? Between areas/urban etc**

We have used the Land Surface Temperature data as provided by the MODIS team. A detailed version of the algorithm is available in Wan (2008).

Wan, Z. 2008. "New Refinements and Validation of the MODIS Land-Surface Temperature/Emissivity Products." *Remote Sensing of Environment* 112 (1): 59–74. doi:10.1016/j.rse.2006.06.026.

**17. L110 be clear that selection of no cloudy days introduces a bias to certain meteorological conditions**

A new sentence has been introduced to be clear about this: "*This introduces a bias in to certain meteorological conditions (clear-sky days), but it is unavoidable.*"

18. L160 cite chapter authors, not the book

The reference has been modified following the suggestion.

**19. Table 1 link to Figure 1 (stations); Define Variance ratio or cite reference**

A link to figure 1 has been introduced in the caption of table 1, as suggested. Also, a new sentence has been added to the caption, to define the scores, including the variance ratio: "*The scores are: Mean bias (model - observed), Root Mean Squared Error (RMSE), and variance ratio (variance of the model divided by variance of the observation)."*

**20. L118 meters -> UK or USA English?**

It has been modified to "metres" to be consistent with UK English.

**21. L170 standard scores or metrics – reference**

The word "standard" has been removed, as there are not "standard" scores in the literature. While the scores we used are very typical, we think it is not fully accurate to describe them as "standard".

**22. L170 2 m**

The sentence has been modified as suggested.

**23. Figure 2 – indicate in the caption where codes for key are explained. Captions should be standalone**

A new sentence has been introduced in the caption following the advice of the referee: "*The* UC-ERA, UC-FC and WRF legend codes are defined in section 2, while OBS is the observation."

24. L185 Check in all places oC; or express in terms of K and remove o. In some places reversed  $\circ$  C (e.g. Table 1)

The temperature units have been checked and corrected throughout all the manuscript.

**25. L189 – Garcia**

The reference has been corrected as suggested.

**26. L193 - Note importance of land cover. What do they represent in terms of Local Climate Zones?**

There are no significant differences in the local climate of the stations, apart from the effect of urbanization. Furthermore, the new figures give much more information about the land uses.

**27. L193 cite these previous studies**

This sentences refers to the (Moreno-Garcia, 1994) paper cited above.

**28. L200 on – But don't you need to check all grids now? Land use? Advection? Etc. what**

Advection of the UHI over the rural location is a plausible hypothesis under certain conditions given that it is close to a urbanized area. This is especially true in WRF, in which, due to its lower resolution, the rural gridpoint is close to a urban gridpoint. However, the advection of the UHI seems to be very small in Barcelona, according to the models, and the sea breeze does not blow from the city to the rural station. But, bearing all these considerations in mind, we must note that the final goal of the study is not a perfect measurement of the UHI in Barcelona, which would require a measurement campaign, but the evaluation of the model. If there is an influence of the UHI in the rural location through advection, this should be simulated by the models.

**29. Figure 3 – be explicit about UHI – temperature difference**

Both in former figures 2 and 3 (which are figures 4 and 5 in the new manuscript), the title of the right panel has been changed from "UHI" to Urban – Rural to be more explicit.

**30. L 222 (e.g. Figure 4) rather than which are here depicted in . . ..**

The sentence has been modified as suggested and it now reads as follows: "*It is interesting to highlight the day-to-day variability of the observed and simulated times series for the month of May (figure 5)*."

**31. L231 Figure 5 (introduce space)**

This mistake has been corrected in new version of the manuscript.

**32. Figure 4 – Significant figures! Relabel X-axis no need for May 2011 on all as in caption**

The figure has been edited as suggested.

33. L234 as above space between number and units

This mistake has been corrected, thanks.

34. L240 as above evaluation rather than validation

This has been corrected through all the manuscript.

35. L 242 – typos near delete .advection

The typo has been corrected.

36. L256 – be more explicit about long spin-up. Some suggest for certain models 10-20 years are needed to get soil moisture characteristics correct.

They suggest this, but do not show the proof.

37. Line 319 data were not was

The sentences has been modified as suggested by the reviewer.

**M. Mohan comment**

Authors want to thank M. Mohan for this comprehensive review.

**General Comments:**

The paper addresses advantages of using a fast Urban Climate Model (UrbClim) over a full scale mesoscale model (WRF) by using a case study depicting UHI simulation for the city of Barcelona. Urban Climate Model is relatively of high resolution that is driven by ERA interim reanalysis data or GFM of ECMWF. The model has earlier been validated for other European cities and now being implemented for Barcelona and utilized for climate projections. The model is demonstrated to be computationally efficient with higher resolution than a mesoscale model. However, as mentioned in the title, the model (UrbClim) is not a urban canopy model as canopy features are not included in the true sense and the mesoscale model WRF does not include WRF- UCM. Hence the title needs to be appropriately modified with direct usage of UrbClim Though UHI is derived from the temperature differences, the paper presntlly lacks the robustness of estimating UHI. Further, model validation shall include meteorological parameters such as Wind speed and PBL etc. to demonstrate the efficacy of the two models. It would be interesting to see the performance of all the key meteorological parameters from the two models to examine the comparable performance and infer of the desired efficiency of UrbClim and this model comparison alone is adequate to make this point.

The comments are elaborated below:

1. Line 90 onwards: Station number 6 is located on a rooftop at 33m while Station number 5 which is a rural station in located in a delta. The thermal properties of land surface and a land use predominently reflecting a water body is drastically different and the fact that no inversion is observed over rural station (considered here as reference station for UHI) would lead to erroneous representation of estimated UHI of the city.

The objection here is unclear. Note that the river forming the delta is quite small, and the water bodies do not prevail over the location of the rural station at all. The land use is dry land, cultivated mainly with cereal but also other species, with artificial irrigation (but not flooded). Also, the sentence mentioning inversion in the paper was confusing, and it has been replaced by: *"The rural station is located in a delta, and therefore the surrounding topography is flat, with no relevant*

*orographic objects between the two stations*". The terrain is also very similar in both rural and urban stations: coastal plain with small or no slope.

2. Figure 1 a and 1 b: Figure 1 b should also depict the rural and urban stations. Both figures should depict coordinates (lat/long) using ArcGIS or another appropriate software. It appears that Urban and rural stations are not within the same nest and with similar resolution. This might affect the results somewhat. This needs to be clarified.

All the maps of the paper include now coordinates in their axis. Also note that the urban and rural stations are in the same nest in the WRF domain. In figure 1b, the black contour represents the UrbClim domain. Thus, it is not possible to plot the stations in figure 1b, because the region covered by the map is too large.

3. Brief description of incorporation of urban canopy model (physics/key eq.) in Urb-Clim model shall be included to explain the science and for comparison with mesoscale model to understand level of simplification.

The details are provided in the references. As the scope the paper is very applied, we think that discussing details about the physics and the parameterizations is out of place.

What are the time step for running these two models? A climate model run needs larger time step and a weather model like WRF requires lower time step to run.

Information about the time steps is now included in the paper. For UrbClim is 20s for the land surface module, but the atmospheric scheme is solved with a variable time step, using the Courant-Friedrichs-Lévy stability criterion to determine the longest possible stable time step.

**If this difference is large between the two models, UrbClim will obviously be computationally much more efficient. It would be more appropriate to compare UrbClim with another Climate model with similar or comparable time step resolution for depicting it's efficiency for predicting average temperatures from May to Sept. 2011. Essentially, this efficiency is one of the major claim by the authors in this paper and that is not shown by considering two dissimilar type of models in terms of phenomenal applicability of UHI. Further by including other meteorological parameters such as wind speed, fluxes and PBL and demonstrating the comparative model performance would prove the point more effectively.**

The new version of the manuscript is clear about the different scope and applicability of WRF and UrbClim, and the goal is not to replace WRF with UrbClim is all applications. UrbClim is focused on reproducing the UHI and the heat stress, so this defines the variables used in the study. WRF is being used in many studies focused on the UHI that could use a faster and simpler model as UrbClim, and our goal is to show that.

Also, note that the main reason of the performance difference is not the time step or the number of gridpoints, but the simplicity of the dynamical core, and that UrbClim does run less parameterizations (there are no cumulus, radiation and microphysics parameterizations). Thus, the paper does not try to compare the computational efficiency of two similar models, but of two very different approaches to simulate the same thing (the UHI of Barcelona).

**4. UHI requires detailed description of measurement sites including station pictures with surroundings.**

Pictures of the measurement sites are available in the web page of the Catalan meteorological

Service:

- urban: http://meteo.cat/observacions/xema/dades?codi=X4&dia=2016-05-26T13:30Z

- rural: http://meteo.cat/observacions/xema/dades?codi=XL&dia=2016-05-26T13:30Z

2 station data will not be sufficient for robustness. Thus, it is suggested that model implementation and efficacy claim is limited to temperature predictions and other met. parameters shall also be included for this purpose.

We disagree with this statement. Two well placed and maintained stations can provide a good representation of the UHI. Furthermore, the study is focused in the UHI, and therefore the evaluation of other meteorological parameters is out of the scope.

Further, authors claim that there are 11 met. stations in the domain; however data for only 2 stations is being used to depict UHI. Were the chosen stations showed the maximum UHI? How about UHI for other stations ? Could they reflect justifiable trends vis-a- vis their LULC? Most of the study would use 25 or more stations for UHI(Mohan et al., 2013:, Assessment of urban heat island effect for different land use–land cover from micrometeorological measurements and remote sensing data for megacity Delhi, Theoretical and Applied Climatology, 112, 647-658. DOI 10.1007/s00704-012-0758-z ).

The other possible station pairs to be used to measure the UHI have different problems: Are too far away, located at very different height above sea level, separated by topographical barriers, too close to the sea, or to the airport. The station pair used was chosen after a careful analysis of all the data.

On line 190, authors mention that " The measurement of the UHI with only two points has some limitations, as it may be sensitive to very local features such as the land use in the vicinity of the stations. However, the representativeness of these points has been carefully checked with high resolution satellite images". No details are provided as to how it has been checked.

They have been checked by means of high resolution google earth images, together with the pictures provided in the web page of the meteorological service (find more details in previous comments).

In addition, the agreement with previous studies increases our confidence in the results here presented. No details of any previous studies provided here.

A previous study is referenced (Moreno-García et al. 1994).

Satellite data would represent LST and not air temperature while UHI estimated and shown are based on air temperature.

Yes, this was already explained in the original version of the manuscript. LST is the only possible way to evaluate the spatial pattern of the simulations.

5. Does the same trend in UHI was seen based on LST also? It is not clear. UHI phenomena occurs on a diurnal scale for which WRF-UCM is applied (Bhati and Mohan, 2015; TAC; doi:10.1007/s00704-015-1589-5).

RMSE of temperature of about 2 deg. is acceptable WMOas per guidelines on shorter time scales

(of an hour or day as demonstrated in this study. Therefore results can be sufficiently robust for temperatures averaged over 5 months but the differences of 2.5 deg. as in UHI may not be ; hence model application seems appropriate for temperatures and not for UHI.

Unfortunately, we are not sure we completely understand what the comment is trying to refer to.

Morerover, different heights of urban and rural stations will further add to this uncertainty.

As mentioned in the original version of the manuscript, the height difference can account for a difference of 0.15-0.25 degrees depending on the gradient considered. Also see the answer to referee #2 regarding the representativeness of temperatures measured in roofs (page 6 of this document).

6. Simulations are carried out from May to Sept, 2011. It is not clear for statistical paired analysis what temporal and spatial resolution and how many data points are used?

This information is now included in the text and in table 2.

7. It will be good to include the monthly variations of UHI considering seasonality in May to Sept. data and examine the trends ? Similarly for the air temperatures and LST as well the monthly variations could be included.

This interesting suggestion from the reviewer will be addressed in a future article, in which much longer simulations, of the order of 30 years, will be performed in order to analyze the climatology of the city. In this article, however, we simply wanted to perform an initial evaluation of the model, for the particular case of the city of Barcelona.

8. Based on model simulations the spatial variation of UHI needs to be studied. It shall be shown whether spatial variation shows maximum difference at the two selected points and other places in the domain are depicting lower UHI so that urban -rural contrast can be deciphered.

Unfortunately, there are no observations of the spatial pattern of the 2m temperature. Figure 7 depicts the spatial pattern found in the simulations, which shows that the two selected stations are representative of the maximum difference between the city centre and the surrounding areas.

9. The title mentions fast Urban Canopy Model while the abstract and text categorises this as Urban Climate Model and the mesoscale model used is WRF and not WRF-UCM. As per the text, WRF includes USGS LULC and no mention of WRF UCM is made. WRF has tremendous scope of improvement by using recent LULC other than USGS and including Urban Canopy as demonstrated by Bhati and Mohan (2015). Thus urban canopy model in the title may be replaced with urban climate model.

The title has been modified in the new version of the manuscript to account for this, and now it reads: "*Advantages of using a fast urban boundary layer model as compared to a full mesoscale model to simulate the urban heat island of Barcelona*." which we believe is more accurate. As mentioned in the original version of the manuscript, the LULC used in all the simulations in the paper is the European CORINE dataset, which is considerably more accurate than the USGS dataset included in WRF.

[revised manuscript text omitted]

---

## Editor Decision (ED1)

In response to reviewer 1 about the LST bias in both URBClim and WRF as compared to MODIS, the explanation you provided to that comment also needs to be in the manuscript, rather than just the reply.

Reviewer 2 described your work as comparing an offline urban canopy model to a fully coupled meso-scale atmospheric model WRF. You disagree in this respect that UrbClim is not an offline model. However, I think you are mis-understanding the point made by the reviewer. In WRF, the inputs to the PBL scheme come from the LSM, SLC, as well as the dynamical core, which provides the prognostic variables. Yes, Urbclim parameterizes the PBL, but the inputs from ERA interim are fixed. This is not the case in WRF where the PBL scheme and the inputs to the PBL scheme have two-way interaction. In this sense, Urbclim is an offline model compared to WRF. You need to better explain the inner-workings of UrbClim and the key differences to WRF to make this clearer (not just refer to the paper which describes Urbclim).

Reviewer 2 asked for more physical explanation of the processes, which you have not carried out. However, since UrbClim also includes a land surface scheme, a comparison of energy balance from UC-ERA with WRF, e.g., sensible and latent heat, may help shed some light. I think you could try a bit harder in this respect. This is still plenty of room in this paper for a bit more analysis of the processes. I do not disagree that it is hard, but you should try nonetheless.

Reviewer 3 made a very good point about the use of statistical downscaling. Your reply needs to be incorporated within the manuscript, perhaps in the introduction.

Title: The main point here is that UrbClim is a "stand-alone" urban boundary layer model which is not coupled to a fully fledged atmospheric model, hence it is fast, which one would infer. I suggest to change the tittle to "Advantages of using a fast stand-alone urban boundary layer model as compared to a fully coupled mesoscale model to simulate ……..".

The abstract should generally be one paragraph.

Abstract, line 3, it is unclear what you mean by "these simulations", as the previous lines do not refer to any simulations. I suggest "by high resolution (sub kilometer) fully coupled land-atmosphere simulations using urban canopy parameterizations" – This is more precise.

Abstract, line 4, "an urban" not "a urban".

Why do you use upper case for "Urban Heat Island" throughout the abstract? Define the UHI acronym the first time, and use it in the rest of the abstract.

Abstract, line 10, as far as commonly used re-analysis products go (NNRP, FNL, ERA40, ERA-Interim, etc), 70 km resolution is pretty much as high as it gets! So your use of the term "relatively low resolution reanalysis (70 km)" is what the

rest of the climate community considers as high-resolution reanalysis. This needs to be changed.

Abstract, lines 11-12, simply change to "In addition, the effect of using driving data from a higher resolution forecast model (15 km) is explored in the case of UrbClim."

Abstract, final paragraph. This is where you describe the main results and this lacks detail. I suggest to add 70 km in brackets after "reanalysis data" at line 14 at the end of the sentence. You need to state the actual "problem with the winds" this is too broad, and "day-to-day correlation" of what? I do not like the use of terms such as "the problem disappears" when referring to issues with models. Rather, "errors are substantially reduced from x units to y units" – is a lot more scientific and useful to a reader, rather than "the problem disappears".

Additionally, provided it fits within the work limit for the abstract, a sentence or two on when it would be appropriate, and more importantly, when it would NOT be appropriate to use UrbClim would be good to mention. This is really important information I would expect in an abstract.

Page 2, line 31, you previously defined the UHI acronym on page 1, yet you use the full term. Please check this through the manuscript and be consistent.

Page 2, line 38, "Others have highlighted" rather than "Other authors", put the reference at the end of the sentence.

You generally have too many one-sentence paragraphs, which makes the paper a bit "jumpy". For example, page 3, lines 63, and 75, there are two paragraphs of one sentence each. You need to improve on the overall structure of your paper in terms of paragraphing.

Page 2, line 49, sentence starting with "Here we show". This is a result and does not belong in an introduction.

Page 2, lines 54 to 56, where you describe the scope of UrbClim versus RCMs. Computational efficiency is only one aspect. You also need to mention that Urbclim, cannot be used to investigate the effect of UHI on atmospheric circulation, such as interactions with the sea breeze and convection/storm initiation, which has been shown by other studies. I think this is very important to make clear, rather than just the computational aspect.

Page 3 ,line 60, wrong cite command for Chen et al. (2011).

The description of the climatology of Barcelona etc, should be in the methods section, not the introduction.

Page 3 , second dot point, "of the UrbClim simulation" rather than "of this run".

Page 5, line 98, what is a "well maintained" station is subjective, rather, is there any form of Quality control applied to this data before you used it? This may be more useful information to provide.

Page 6, line 131, "a minimum" rather than "the minimum".

Page 6, line 134- "The boundary data needs to be read from a lower resolution model" – This is too broad and not detailed enough. State exactly what inputs UrbClim requires. Your use of the term "boundary data" is rather broad as well. WRF, as any RCM, uses input data at the lateral boundaries, and the influence of this data decreases within the sponge zone. I suspect UrbClim does not have an actual "boundary zone" as RCMs do?

Page 6, line 136, you state "Mesoscale models are tied to their driving models by the boundary conditions. Yet, they develop internal variability". To me, this is stating the obvious, of course they will develop internal variability as the influence of the input boundary conditions decreases exponentially within the relaxation zone, and of course the driving data has a large influence.

Page 6, line 140, not just "wind" but as a consequence the advection of heat and moisture?

Page 6, line 141 to 143, I think the point you should make here, is that conventional RCMs have to use several nests to achieve high resolutions of 1 km or less, whereas this is not an issue with UrbClim.

Page 6, line 151, wrong cite command for Ridder and Schayes, (1997) and De Ridder et al. (2015). Please PROOF READ your manuscript.

Page 6, line 160. Again here, poor paragraph structure, you cannot start a paragraph with "In contrast", it does not flow.

Page 7, line 178, it's "Forecasting" and Not "Forecast".
 Page 8, line 196, "previous studies" rather than "previous works".

Table 1  shows UC-ERA has consistently larger +ve biases and RMSE as compared to WRF across all stations. This is an important result I would expect to find in the abstract.

Page 9, line 205, "at some stations", not "in some stations".

Page 9, line 206, rather than "see below", use "this is discussed later in the manuscript" or something along those lines.

Page 9, line 208, you state "Instead, UC-FC and WRF show similar, smaller scores, which indicate the good performance of these simulations". You are stating the obvious here, of course smaller errors mean improved performance, one can assume the reader will make this connection. Rather, the point here is that UrbClim is very sensitive to the driving data, and you need driving data at 15 km

for the errors to be the same order of magnitude as WRF. This is significant as 15 km driving data is not routinely available over long time periods! This is far more important to discuss.

Page 10, line 214, wrong cite command again. Line 219, "at both stations" not "in both stations".

Your supplementary figures have no captions, so I do not know what I have looking it.

Page 12, line 250, I don't really see the point of comparing MAE of 0.8 to 1.11, this is a difference of 0.3C, rather small. UC-FC is only marginally better than WRF.

Page 12, line 22, replace "more biased" with "has slightly larger biases" .

A number of studies have compared different PBL schemes in WRF against wind speed observations from Atmospheric soundings. You should actually reference these here and look at the RMSE, MAE and biases they report. I largely suspect that using different PBL and other schemes in WRF would result in changes in biases compared to observations which are larger than the differences you find between WRF and UC-FC, which would imply it would be entirely plausible that different configuration(s) of WRF could easily result in even larger or smaller errors than UC-FC. We cannot tell unless we do the simulations, but you should discuss this in more detail on page 12.

Page 13, below Fig 8 – again here, a one-sentence paragraph which does not really flow.

Page 13, lines 285-286 – You refer to biases in Figure 8, but figure 8 is not a bias plot? Not sure I follow here.

Page 15, the first dot point is not really a conclusion of your study as the aim was not to quantify the UHI of Barcelona. This should be removed as a conclusion.

Second dot point – What systematic biases? State them!

Third dot point – Of course WRF will provide less detailed spatial information, You ran it at a coarser resolution as compared to UC-ERA! This is to be expected and not a conclusion of the study. You fail to mention that UC-ERA had consistently larger biases than WRF, which is far more important, as well as the fact that UrbClim needs inputs at considerably higher resolution than routine available, i.e., 15 km, to really show a distinction from WRF. This is far more important.

In your conclusion, you state that this opens the door to running UrbClim with GCMs simulations of future climate. There are several problems here. Firstly, GCMs have much coarser resolution than 70 km, and you have clearly shown the resolution of input data has a large influence on UrbClim. Most GCMs have

greater than 150 km resolution. Secondly, you ran UrbClim with a re-analysis, which is completely different to GCM simulations of current and future climate. You are extrapolating too much here.

It would be much more useful to have a paragraph which objectively discusses, in which circumstances on should choose UrbClim over an RCM such as WRF, and what the user should be mindful of, rather than trying to extrapolate too much. Some of this needs to be reflected in the abstract.

Finally, in section 3.3, you provide a lot of detail about compilers etc. I do not think this add much value at all to the paper. One expects that a stand-alone model such as UrbClim would require significantly less computational resources than an RCM such as WRF. You can simply state that given your WRF setup, UrbClim was about 130 times faster than WRF. All this detail on the OS, compiler verisons, MPI stuff etc is not really adding much. This detail is only relevant when comparing slightly different versions of the same code, rather than two completely different codes. It would add more value to the paper, to spend more words on physical processes.

---

## Author Response (AR2)

Dear Jatin

The authors would like to thank you for the review, that lead to further improving the manuscript. In general, we see that your seem to think that the paper is not fair with WRF. We acknowledge that the first submitted version had some inaccuracies and ambiguities in this sense, but we believe that these have been fully removed during the two reviews. Please see the comments below.

Best regards

The authors

In response to reviewer 1 about the LST bias in both URBClim and WRF as compared to MODIS, the explanation you provided to that comment also needs to be in the manuscript, rather than just the reply.

The first two sentences of the explanation provided to the reviewer have now been included in the methods section. Emissivity was already mentioned in the results section, but without giving any supporting quantities: *"Determination of emissivity over urban areas is notoriously difficult and subject to large uncertainties, which could explain at least part of model deviation of LST."* Given the difficulty of the problem and that we are not experts in teledetection, we think that it is better to leave this sentence as it is.

Reviewer 2 described your work as comparing an offline urban canopy model to a fully coupled meso-scale atmospheric model WRF. You disagree in this respect that UrbClim is not an offline model. However, I think you are mis-understanding the point made by the reviewer. In WRF, the inputs to the PBL scheme come from the LSM, SLC, as well as the dynamical core, which provides the prognostic variables. Yes, Urbclim parameterizes the PBL, but the inputs from ERA interim are fixed. This is not the case in WRF where the PBL scheme and the inputs to the PBL scheme have two-way interaction. In this sense, Urbclim is an offline model compared to WRF. You need to better explain the inner-workings of UrbClim and the key differences to WRF to make this clearer (not just refer to the paper which describes Urbclim).

We disagree with your appreciation. As we see it, the definition of what an "offline" model is is unclear here. Neither WRF or UrbClim provide feedback to ERA-Interim. The "offline" term usually refers to isolated parameterizations that run with fixed input data and we are not working under these premises. UrbClim is more complex than that as it does have a simple prognostic equation for the lower 3 km, plus coupled land surface and PBL.

We think that the UrbClim description provided is detailed enough, and it has been extended a bit in the present version of the manuscript. We believe that the different scope of WRF and UrbClim is now clear enough throughout the text.

Reviewer 2 asked for more physical explanation of the processes, which you have not carried out. However, since UrbClim also includes a land surface scheme, a comparison of energy balance from UC-ERA with WRF, e.g., sensible and latent heat, may help shed some light. I think you could try a bit harder in this respect. This is still plenty of room in this paper for a bit more analysis of the processes. I do not disagree that it is hard, but you should try nonetheless.

This would be very interesting. Unfortunately, is beyond the scope of the current study, and to easy to address given the lack of adequate observations to use as reference. However, we woul like to stress that the former has been partially been addressed in De Ridder et al. (2015) and Lauwaet et al. (2016).

Reviewer 3 made a very good point about the use of statistical downscaling. Your reply needs to be incorporated within the manuscript, perhaps in the introduction.

As suggested, in the new version of the manuscript part of the answer to #reviewer3 has been included in the introduction.

*"Statistical downscaling can also be considered as an alternative (os complementary) methodology to asses the urban climate. However, statistical downscaling is observation-dependent, and spatially detailed observations at a city scale covering long periods are very rare. UrbClim, instead, allows for long multi-decadal adaptation experiments where changes in the city surface parameters can be tested (e.g., the color of the roofs or the evaluation of the effects that changes in construction materials in building may have). The main aim of this study simulate only the fundamental processes that cause the UHI, so the model is lightweight, but still based on physics, and thus allowing sensitivity experiments to be conducted."*

Title: The main point here is that UrbClim is a "stand-alone" urban boundary layer model which is not coupled to a fully fledged atmospheric model, hence it is fast, which one would infer. I suggest to change the tittle to "Advantages of using a fast stand-alone urban boundary layer model as compared to a fully coupled mesoscale model to simulate .......".

As explained above, the meaning of "offline" or "stand-alone" is unclear and we do not feel comfortable either with the suggested changes. We think that "urban boundary layer model" is a good definition of UrbClim, as formerly stated in De Ridder et al. (2015) and Lauwaet et al. (2016).

The abstract should generally be one paragraph.

In the new version of the manuscript, the abstract is only one paragraph.

Abstract, line 3, it is unclear what you mean by "these simulations", as the previous lines do not refer to any simulations. I suggest "by high resolution (sub kilometer) fully coupled land-atmosphere simulations using urban canopy parameterizations" – This is more precise.

The sentence has been modified as suggested.

Abstract, line 4, "an urban" not "a urban".

The mistake has been corrected.

Why do you use upper case for "Urban Heat Island" throughout the abstract? Define the UHI acronym the first time, and use it in the rest of the abstract.

The text has been modified as suggested.

Abstract, line 10, as far as commonly used re-analysis products go (NNRP, FNL,

ERA40, ERA-Interim, etc), 70 km resolution is pretty much as high as it gets! So your use of the term "relatively low resolution reanalysis (70 km)" is what the rest of the climate community considers as high-resolution reanalysis. This needs to be changed.

The sentence has been modified as suggested, and now it reads as follows: *"This comparison is performed with driving data from ERA-Interim reanalysis data (70 km)"*.

Abstract, lines 11-12, simply change to "In addition, the effect of using driving data from a higher resolution forecast model (15 km) is explored in the case of UrbClim."

The sentence has been changed as suggested.

Abstract, final paragraph. This is where you describe the main results and this lacks detail. I suggest to add 70 km in brackets after "reanalysis data" at line 14 at the end of the sentence. You need to state the actual "problem with the winds" this is too broad, and "day-to-day correlation" of what? I do not like the use of terms such as "the problem disappears" when referring to issues with models. Rather, "errors are substantially reduced from x units to y units" – is a lot more scientific and useful to a reader, rather than "the problem disappears".

Additionally, provided it fits within the work limit for the abstract, a sentence or two on when it would be appropriate, and more importantly, when it would NOT be appropriate to use UrbClim would be good to mention. This is really important information I would expect in an abstract.

We agree in that this part lacks detail. The final part of the abstract has been re-written to accommodate your suggestions:

*"The results show that the performance on reproducing the average UHI in the simple model is generally comparable to the one in the mesoscale model when driven with reanalysis data (70 km). However, the simple model needs higher resolution data from the forecast model (15 km) to correctly reproduce the variability of the UHI at a daily scale, which is related to the wind speed. We conclude that medium-complexity models as UrbClim are a suitable tool to simulate the urban climate, but that they are sensitive to the ability of the input data to represent the local wind regime. UrbClim is well suited to develop impact and adaptation studies at a city scale without high computing requirements, and mesoscale models are yet required to study the two-way interaction between the city and the atmosphere."*

Page 2, line 31, you previously defined the UHI acronym on page 1, yet you use the full term. Please check this through the manuscript and be consistent.

As requested, the mentions to Urban Heat Island have been replaced with UHI here and when necessary to make the text consistent.

Page 2, line 38, "Others have highlighted" rather than "Other authors", put the reference at the end of the sentence.

The sentence has been modified as suggested.

You generally have too many one-sentence paragraphs, which makes the paper a bit "jumpy". For example, page 3, lines 63, and 75, there are two paragraphs of one sentence each. You need to improve on the overall structure of your paper in terms of paragraphing.

The paragraphing of the manuscript has been revised as requested.

Page 2, line 49, sentence starting with "Here we show". This is a result and does not belong in an introduction.

The sentence has been modified so it does not advance the results: *"Here we study, by using a simplified model that only accounts for the Planetary Boundary Layer (PBL) and the surface physics, the possibility to reach resolutions of 250 m with affordable computational resources".*

Page 2, lines 54 to 56, where you describe the scope of UrbClim versus RCMs. Computational efficiency is only one aspect. You also need to mention that Urbclim, cannot be used to investigate the effect of UHI on atmospheric circulation, such as interactions with the sea breeze and convection/storm initiation, which has been shown by other studies. I think this is very important to make clear, rather than just the computational aspect.

A new sentence has been introduced in the introduction to account for this: *"While this scope covers many applications, it must be mentioned that RCMs are required to reproduce the two-way interaction between the city and the atmosphere affecting the rain, storm initiation and other phenomena, at the expense of a much larger computational cost."*

Page 3 ,line 60, wrong cite command for Chen et al. (2011).

This mistake has been corrected.

The description of the climatology of Barcelona etc, should be in the methods section, not the introduction.

This paragraph describes the climatology of Barcelona in the context of heat and heat waves hazard and climate change, to explain how it is a especially vulnerable city, which adds relevance to the study. In this sense, we think that the introduction is the proper place for this paragraph.

Page 3 , second dot point, "of the UrbClim simulation" rather than "of this run".

The sentence has been modified as suggested.

Page 5, line 98, what is a "well maintained" station is subjective, rather, is there any form of Quality control applied to this data before you used it? This may be more useful information to provide.

Yes, there is a routine quality control carried out by the providers. Additionally, we also inspected the data for outliers and bad quality data. This is now mentioned in the text: *"Both data providers carry on a quality control to this data before distributing it".*

Page 6, line 131, "a minimum" rather than "the minimum".

The sentence has been corrected as suggested.

Page 6, line 134- "The boundary data needs to be read from a lower resolution model" – This is too broad and not detailed enough. State exactly what inputs UrbClim requires. Your use of the term "boundary data" is rather broad as well. WRF, as any RCM, uses input data at the lateral boundaries, and the influence of this data decreases within the sponge zone. I suspect UrbClim does not have an actual "boundary zone" as RCMs do?

The boundary condition scheme is different in UrbClim than in most of the RCMs, and it is described in detail in De Ridder et al. (2015). This description has been summarized in the manuscript as follows: "Apart *from the variables usually fed to the mesoscale models, UrbClim needs also the radiative fluxes and precipitation. Instead using a relaxation zone, UrbClim imposes the driving model data in the inflow boundary points, and a "zero gradient" condition, that lets the perturbations flow outside the domain, in the outflow points.*"

Page 6, line 136, you state "Mesoscale models are tied to their driving models by the boundary conditions. Yet, they develop internal variability". To me, this is stating the obvious, of course they will develop internal variability as the influence of the input boundary conditions decreases exponentially within the relaxation zone, and of course the driving data has a large influence.

The sentence is more intended to serve as a definition of the term internal variability than to state nothing new.

Page 6, line 140, not just "wind" but as a consequence the advection of heat and moisture?

Page 6, line 141 to 143, I think the point you should make here, is that conventional RCMs have to use several nests to achieve high resolutions of 1 km or less, whereas this is not an issue with UrbClim.

We agree in that this part (the lines mentioned by the last two comments) is not clear enough and, as we think it is essential for the interpretation of the results, it has been carefully re-written:

*"Mesoscale models develop their own variability and structures respect to the lower resolution models driving them, which is called internal variability \citep{giorgi_2000}. By design, UrbClim does produce a significantly smaller internal variability than a mesoscale model. This can be considered as a disadvantage, as it does not permit studying the full two-way interaction of the city with all the regional troposphere. But, on the other hand, it is the key for saving computational power. As the UHI is rooted in the surface properties and the heat storage in the ground, using a model like UrbClim is reasonable. This model does work approximately as a wind tunnel, without creating regional structures in the atmospheric flow, so it is possible to nest it directly in much lower resolution models without creating intermediate nests. Nonetheless, this resolution jump can affect the quality of the simulation if the driving model does not accurately reproduce the local climate. Mesoscale models need these intermediate nests for the inconsistencies between internal variability and driving data not to blow up the numerical stability. This trade-off between the internal variability and the computational efficiency will be key for the interpretation of the results in this study."*

*Page 6, line 151, wrong cite command for Ridder and Schayes, (1997) and De Ridder et al. (2015). Please PROOF READ your manuscript.*

The mistake has been corrected.

*Page 6, line 160. Again here, poor paragraph structure, you cannot start a paragraph with "In contrast", it does not flow.*

The paragraph starting with "in contrast" has been merged with the previous one.

*Page 7, line 178, it's "Forecasting" and Not "Forecast".*
 *Page 8, line 196, "previous studies" rather than "previous works".*

Both mistakes have been corrected in the new version of the manuscript.

*Table 1 shows UC-ERA has consistently larger +ve biases and RMSE as compared to WRF across all stations. This is an important result I would expect to find in the abstract.*
The warm biases in UC-ERA are very related to the lack of sea breeze in this run, as explained in the text. This is now also mentioned in the abstract: *"This lack of accuracy on reproducing the wind speed, especially the sea breeze daily cycle, which is strong in Barcelona, causes also a warm bias in the reanalysis driven UrbClim run".*

*Page 9, line 205, "at some stations", not "in some stations".*

The mistake has been corrected.

*Page 9, line 206, rather than "see below", use "this is discussed later in the manuscript" or something along those lines.*

The sentence has been changed as suggested.

*Page 9, line 208, you state "Instead, UC-FC and WRF show similar, smaller scores, which indicate the good performance of these simulations". You are stating the obvious here, of course smaller errors mean improved performance, one can assume the reader will make this connection. Rather, the point here is that UrbClim is very sensitive to the driving data, and you need driving data at 15 km for the errors to be the same order of magnitude as WRF. This is significant as 15 km driving data is not routinely available over long time periods! This is far more important to discuss.*

Please note that this is discussed afterwards (see line 277 and forward). Also note that this result is specific for Barcelona, as many other cities do not have this sea breeze problem. The 15 km data can also come from a regional reanalysis or from a RCM, and yet it would be significantly cheaper than to use WRF up to 250 m horizontal resolution. Soon new ERA5 reanalyses will be available with 32 km resolution and they will be likely sufficient enough for driving UrbClim even in more cities that ERA-Interim. In this regard, we think that UrbClim results are an impressive achievement.

*Page 10, line 214, wrong cite command again. Line 219, "at both stations" not "in both stations".*

The mistakes mentioned have been corrected in the text.

Your supplementary figures have no captions, so I do not know what I have looking it.

The supplementary figures are now arranged in a single pdf with captions.

Page 12, line 250, I don't really see the point of comparing MAE of 0.8 to 1.11, this is a difference of 0.3C, rather small. UC-FC is only marginally better than WRF.

In our opinion, on the long term, a 0.3ºC difference in the MAE is not necessarily small. In this case, the difference between UC-ERA and UC-FC is even smaller, but UC-ERA has large occasional errors which can be relevant for a user, and these are reflected in the MAE difference. In the new version of the manuscript we mention that WRF has a small but persistent error in the daily urban-rural difference: *"This underestimation is small, albeit persistent."*

Page 12, line 22, replace "more biased" with "has slightly larger biases" .

We have considered this and are not going to apply this proposed change to the manuscript. In figure 4 WRF is clearly more biased than UC-FC (not UC-ERA), with up to -2ºC in the urban location during the day. As a consequence, we don't consider that "slightly" is the proper word to describe this difference.

A number of studies have compared different PBL schemes in WRF against wind speed observations from Atmospheric soundings. You should actually reference these here and look at the RMSE, MAE and biases they report. I largely suspect that using different PBL and other schemes in WRF would result in changes in biases compared to observations which are larger than the differences you find between WRF and UC-FC, which would imply it would be entirely plausible that different configuration(s) of WRF could easily result in even larger or smaller errors than UC-FC. We cannot tell unless we do the simulations, but you should discuss this in more detail on page 12.

In the last submitted version of the manuscript there is a mention to this: *"As WRF is very customizable, it could be possible, in principle, to find a configuration that removes these biases."* However, as you state, without doing more simulations is difficult to go further in this direction. The WRF configuration used for the paper is not random, as we used the more popular and reliable parameterizations (Yonsei University Scheme in the case of PBL). Anyway, the goal of the paper is not a comprehensive evaluation of WRF, which we do not criticize, but to compare a WRF run with UrbClim to discuss their skill as well as their limitations.

Page 13, below Fig 8 – again here, a one-sentence paragraph which does not really flow.

The paragraph has been merged with the previous one.

Page 13, lines 285-286 – You refer to biases in Figure 8, but figure 8 is not a bias plot? Not sure I follow here.

Figure 8 displays maps with modeled averages and the observation. Thus, despite it is not a bias plot per se, the bias can be clearly appreciated, especially thanks to the strong contrast of the

colorbar. We chose this kind of plot in order to show the spatial patterns of the absolute fields, and not only the difference between the two fields as in a bias plot. The sentence has been modified to clarify this:

*"By comparing modeled (panels b and c) and observed (panel a) maps in figure 8, it can be seen that the bias outside the urban areas is found to be relatively small in WRF, and slightly negative in UC-ERA, while the spatial patterns are reasonably similar between the models and the satellite data."*

Page 15, the first dot point is not really a conclusion of your study as the aim was not to quantify the UHI of Barcelona. This should be removed as a conclusion.

While it is not the main goal of the paper, we believe that the result is interesting enough to stress it as a main conclusion, due to the lack of modern modeling studies of this length in the city of Barcelona. Therefore, we advocate decided to keep it as is.

Second dot point – What systematic biases? State them!

As suggested, the warm bias detected in UC-ERA is now mentioned here.

Third dot point – Of course WRF will provide less detailed spatial information, You ran it at a coarser resolution as compared to UC-ERA! This is to be expected and not a conclusion of the study. You fail to mention that UC-ERA had consistently larger biases than WRF, which is far more important, as well as the fact that UrbClim needs inputs at considerably higher resolution than routine available, i.e., 15 km, to really show a distinction from WRF. This is far more important.

This has been corrected in the present version of the manuscript. This point now reads: *"WRF is less biased than the UrbClim run nested in ERA-Interim, and both runs show comparable skill on reproducing the UHI."*

In your conclusion, you state that this opens the door to running UrbClim with GCMs simulations of future climate. There are several problems here. Firstly, GCMs have much coarser resolution than 70 km, and you have clearly shown the resolution of input data has a large influence on UrbClim. Most GCMs have greater than 150 km resolution. Secondly, you ran UrbClim with a re-analysis, which is completely different to GCM simulations of current and future climate. You are extrapolating too much here.

Please note that is not an hypothesis, as it has already been achieved for some cities as part of the RAMSES project, see Lauwaet et al. (2015). It is not easy, but neither it is easy tu run WRF nested in GCMs (in addition of being extremely demanding computationally).

It would be much more useful to have a paragraph which objectively discusses, in which circumstances on should choose UrbClim over an RCM such as WRF, and what the user should be mindful of, rather than trying to extrapolate too much. Some of this needs to be reflected in the abstract.

This is was already discussed in the submitted manuscript: *"The choice between UrbClim and WRF for the simulation of the urban environment largely depends on the type of variable and process*

*that is to be analysed. WRF has the advantage of providing a more detailed and complete description of atmospheric winds and rainfall, which is required in some applications (e.g. pollutant dispersion, urban effect in rainfall). On the other hand, UrbClim has been proven to be as accurate as WRF on reproducing the UHI of Barcelona during the warm season, and several orders of magnitude faster."*

It the present version, it is also mentioned in the abstract: *"UrbClim is well suited to develop impact and adaptation studies at a city scale without high computing requirements, and mesoscale models are yet required to study the two-way interaction between the city and the atmosphere."*

Finally, in section 3.3, you provide a lot of detail about compilers etc. I do not think this add much value at all to the paper. One expects that a stand-alone model such as UrbClim would require significantly less computational resources than an RCM such as WRF. You can simply state that given your WRF setup, UrbClim was about 130 times faster than WRF. All this detail on the OS, compiler verisons, MPI stuff etc is not really adding much. This detail is only relevant when comparing slightly different versions of the same code, rather than two completely different codes. It would add more value to the paper, to spend more words on physical processes.

In our opinion, these details are important to improve reproducibility of our results, and to exactly quantify how much faster than WRF is UrbClim in a systematic way, for the purposes and with the configurations presented, as this may greatly depend on the hardware and compiler used.

**References**

[revised manuscript text omitted]

---

## Author Response (AR3)

Dear Jatin,

We agree that the sentence that you propose is clearer. It has been replaced in the abstract as you suggested, except we changed "well a suited" for "a well suited". Thank you very much for your all your work in the review process.

Best regards,

The authors

Dear Markel et al,

Thanks for considering my comments. I am happy to "agree to disagree" on certain points. The paper makes a valuable contribution in helping scientists in choosing which tool to use when focusing on the UHI. However, the last sentence in your abstract still reads a bit odd to me:

"UrbClim is well suited to develop impact and adaptation studies at a city scale without high computing requirements, and mesoscale models are yet required to study the two-way interaction between the city and the atmosphere."

I suggest rewording to:

"UrbClim is well a suited model for impact and adaptation studies at a city scale without high computing requirements."

The last part of the sentense is ambiguous to me: "and mesoscale models are yet required to study the two-way interaction between the city and the atmosphere"

What do you mean? by "yet required", do you mean "still required"? I'd suggest to get rid of the last part of the sentence or make the meaning more clear, for example:

[revised manuscript text omitted]